# Phylogenomic, structural, and cell biological analyses reveal that *Stenotrophomonas maltophilia* replicates in acidified Rab7A-positive vacuoles of *Acanthamoeba castellanii*

Javier Rivera,[1] Julio C. Valerdi-Negreros,[1,2] Diana M. Vázquez-Enciso,[1,3] Fulvia-Stefany Argueta-Zepeda,[1,3] Pablo Vinuesa[1]

**ABSTRACT** *Acanthamoeba* species are clinically relevant free-living amoebae (FLA) ubiquitously found in soil and water bodies. Metabolically active trophozoites graze on diverse microbes via phagocytosis. However, functional studies on Rab GTPases (Rabs), which are critical for controlling vesicle trafficking and maturation, are scarce for this FLA. This knowledge gap can be partly explained by the limited genetic tools available for *Acanthamoeba* cell biology. Here, we developed plasmids to generate fusions of *A. castellanii* strain Neff proteins to the N- or C-termini of mEGFP and mCherry2. Phylogenomic and structural analyses of the 11 Neff Rab7 paralogs found in the RefSeq assembly revealed that eight of them had non-canonical sequences. After correcting the gene annotation for the Rab7A ortholog, we generated a line stably expressing an mEGFP-Rab7A fusion, demonstrating its correct localization to acidified macropinocytic and phagocytic vacuoles using fluorescence microscopy live cell imaging (LCI). Direct labeling of live *Stenotrophomonas maltophilia* ESTM1D_MKCAZ16_6a (Sm18) cells with pHrodo Red, a pH-sensitive dye, demonstrated that they reside within acidified, Rab7A-positive vacuoles. We constructed new mini-Tn7 delivery plasmids and tagged Sm18 with constitutively expressed mScarlet-I. Co-culture experiments of Neff trophozoites with Sm18::mTn7TC1_Pc_mScarlet-I, coupled with LCI and microplate reader assays, demonstrated that Sm18 underwent multiple replication rounds before reaching the extracellular medium via non-lytic exocytosis. We conclude that *S. maltophilia* belongs to the class of bacteria that can use amoeba as an intracellular replication niche within a *Stenotrophomonas*-containing vacuole that interacts extensively with the endocytic pathway.

**IMPORTANCE** Diverse *Acanthamoeba* lineages (genotypes) are of increasing clinical concern, mainly causing amoebic keratitis and granulomatous amebic encephalitis among other infections. *S. maltophilia* ranks among the top 10 most prevalent multi-drug-resistant opportunistic nosocomial pathogens and is a recurrent member of the microbiome hosted by *Acanthamoeba* and other free-living amoebae. However, little is known about the molecular strategies deployed by *Stenotrophomonas* for an intracellular lifestyle in amoebae and other professional phagocytes such as macrophages, which allow the bacterium to evade the immune system and the action of antibiotics. Our plasmids and easy-to-use microtiter plate co-culture assays should facilitate investigations into the cellular microbiology of *Acanthamoeba* interactions with *Stenotrophomonas* and other opportunistic pathogens, which may ultimately lead to the discovery of new molecular targets and antimicrobial therapies to combat difficult-to-treat infections caused by these ubiquitous microbes.

Address correspondence to Pablo Vinuesa, vinuesa@ccg.unam.mx.

Javier Rivera and Pablo Vinuesa contributed equally to this article. Author order was determined by increasing seniority.

The authors declare no conflict of interest.

See the funding table on p. 25.

**KEYWORDS** *Acanthamoeba*, expression plasmids, small Rab GTPases, endocytic pathway, phagocytosis, vacuole acidification, intracellular replication, cellular microbiology, *Stenotrophomonas maltophilia*, mini-Tn7 delivery plasmids, phylogenomics

Free-living amoebae (FLA) of the genus *Acanthamoeba* (Amoebozoa) have been isolated from soil and aquatic habitats worldwide (1–3). The genus currently includes approximately 30 named species and over 20 so-called genotypes with ill-defined taxonomic status (4–6). *A. castellanii* strain Neff (ATCC 30010) was isolated from soil in California by Robert J. Neff in 1957 (7). Neff cultivated this strain axenically, promoting its early adoption as a model organism for cellular, molecular, and genetic studies (8–10). Metabolically active *Acanthamoeba* trophozoites graze on diverse microbes via phagocytosis. Under unfavorable conditions, trophozoites differentiate into highly resistant dormant cysts that allow cells to endure long periods of famine and various environmental stressors (11, 12). Several *Acanthamoeba* genotypes, particularly T4 sublineages, are of clinical concern (13, 14). Amoebic keratitis is the most prevalent *Acanthamoeba* infection (15) and is primarily contracted via contaminated contact lenses (16). At a lower frequency, this organism can cause granulomatous amebic encephalitis, *Acanthamoeba* pneumonia, and disseminated acanthamoebiasis (14, 17). Wild *Acanthamoeba* cells usually host an intracellular microbiome of digestion-resistant microbes including giant viruses (18, 19). This community may contain diverse endosymbionts (20–22) and other bacteria, some of which, such as *Legionella* and *Vibrio,* can use *Acanthamoeba* as a secondary niche for replication and survival under unfavorable conditions (20, 23). Therefore, *Acanthamoeba* and other FLA act as reservoirs for multiple opportunistic bacterial pathogens, contributing to their dispersal and survival in diverse environments and the selection of virulence-related traits (2, 18, 19, 24–26). Most phagocytosed microbes are digested in phagolysosomes, but a steadily growing list of opportunistic pathogenic bacteria are known to escape this fatal fate in various protozoa, including amoebozoa and other FLA (18, 19, 24, 25). These amoeba-resistant microbes deploy a plethora of vesicular trafficking manipulation strategies to prevent fusion of pathogen-containing vacuoles (PCVs) with lysosomes. Membrane trafficking subversion strategies are multifactorial, species specific, and unknown for most bacterial species hosted by *Acanthamoeba* PCVs (18, 20, 27–29).

Compared with the extensive body of literature available on cellular biology for the model social amoeba *Dictyostelium discoideum* (30, 31), limited knowledge exists on *Acanthamoeba* cell biology. Among the protein families with prominent roles in membrane trafficking, small Rab GTPases (Ras-like proteins from the brain) (32, 33) are critical for controlling vesicular flux and maturation along the endocytic, recycling, and secretory pathways (34). Rabs are an ancient and highly diversified family of the Ras GTPase superfamily (35). Human cells express more than 60 Rab family members, whereas yeast expresses 12 (36). This difference results from lineage-specific gains, losses, and expansions and demonstrates the existence of a core of approximately 12 well-conserved Rab proteins (Rab1, Rab2, Rab4, Rab5, Rab6, Rab7, Rab8, Rab11, Rab18, Rab21, Rab23, and Rab28) in eukaryotes (37). The Rab repertoire of *A. castellanii* strain Neff was studied in the context of phylogenomic and evolutionary analyses, and it was estimated that it encodes 93 Rabs (38, 39). However, as detailed in the discussion, only two publications exist on the functional characterization of *Acanthamoeba* Rab GTPases (40, 41).

Here, we focused on Rab7 because of its prominent role in phagosome maturation and subsequent lysosomal delivery (42–44). We performed a detailed phylogenomic and structural analysis of the Rab7 paralogs found in the *A. castellanii* RefSeq (GCF_000313135.1) assembly and proteome (45), which revealed that 8 (73%) of the 11 Rab7 homologs were non-canonical small Rab GTPases and allowed us to identify the *bona fide* Rab7A ortholog after correcting its gene annotation. Modern cell biology studies aimed at characterizing Rab GTPases typically use fusion protein technology coupled with fluorescence microscopy to analyze their temporal and spatial expression

patterns (46) and to identify the cellular compartments that they associate with (42, 43, 47). However, only the plasmid pGAPDH-EGFP, published by Bateman (48) in 2010, has been regularly used in research involving fusion protein technology for *Acanthamoeba* (49–51). This plasmid has several shortcomings: its sequence is not publicly available, and the proteins of interest can only be fused to the C-terminus of EGFP using a minimal number of restriction sites for cloning.

In this study, we constructed four new plasmids with known sequences designed to generate N- or C-terminal fusions of *Acanthamoeba* proteins to either mEGFP (52) or mCherry2 (51) based on empty backbone mammalian expression plasmids. We replaced the viral promoters driving the expression of *nptII* and fluorescent proteins with *Acanthamoeba* TATA-binding protein (TBP) (53) and glyceraldehyde-3-phosphate dehydrogenase (GAPDH) promoters (48), respectively. We used these plasmids to construct two fluorescent cell lines of *A. castellanii* strain Neff. We showed that the mEGFP-Rab7a fusion labeled endocytic vesicles loaded with fluorescent dextran conjugates, a standard fluid phase tracer for endocytosis. Phagocytosis of pH-sensitive, red-fluorescent pHrodo BioParticles demonstrated that *Acanthamoeba* phagosomes are Rab7A positive, acidic compartments.

We used the Neff:mEGFP-rab7A line to study its interaction with the environmental *Stenotrophomonas maltophilia* isolate ESTM1D_MKCAZ16_6a (Sm18) (54) and gained fundamental cell biological insights into the nature of the *S. maltophilia*-containing vacuole (SmCV). *S. maltophilia* currently ranks among the top 10 most prevalent nosocomial pathogens (55) and has been reported to be a digestion-resistant bacterium isolated from diverse FLA, such as *Vermamoeba* (56) and *Acanthamoeba* (57–61). Both organisms coexist in diverse ecosystems, including drinking water distribution systems (25). Cases of combined *Acanthamoeba* and *Stenotrophomonas* keratitis have been reported (62, 63), demonstrating the health risks posed by these organisms in community settings. However, little is known about the molecular mechanisms underlying intracellular survival and replication of *S. maltophilia* (64–66).

To enable visualization of the *Acanthamoeba-Stenotrophomonas* interaction by live cell imaging (LCI), we developed new mini-Tn7-based vectors to stably tag *S. maltophilia* with mScarlet-I (67) constitutively expressed from the strong Pc promoter of class 1 integrons (68). We performed various co-culture experiments between *A. castellanii* and its mEGFP-Rab7A derivative with Sm18 or Sm18::mTn7-Pc_mScarleI cells stained or not with pHrodo to study the essential aspects of the cellular microbiology of *Acanthamoeba-Stenotrophomonas* interactions. These experiments demonstrated that *S. maltophilia* inhabits acidified Rab7A-positive phagosomes. We complemented the epifluorescence LCI studies with microtiter plate assays that provided kinetic data on SmCV acidification and intracellular replication of *S. maltophilia* in *Acanthamoeba* trophozoites. We conclude that *S. maltophilia* belongs to the class of microbes that uses amoeba as an alternative replication niche within an SmCV that interacts extensively with the endocytic pathway.

## RESULTS

### Construction of new *Acanthamoeba* expression vectors

To facilitate cell biology research on *Acanthamoeba*, we developed four new expression plasmids with known sequences designed to construct N- or C-terminal fusions of *Acanthamoeba* proteins with either mEGFP or mCherry2. To build these plasmids, we modified the popular empty backbone mammalian expression plasmids mEGFP-C1, mEGFP-N1, mCherry2-C1, and mCherry2-N1, a gift from Michael Davidson (AddGene plasmids 54759, 54767 54563, and 54517; Table 1). We replaced the viral promoters driving the expression of the *nptII* gene (conferring neomycin and G418 resistance) and the fluorescent proteins (FPs) with the *Acanthamoeba* TBP (69) and GAPDH (48) promoters, respectively, as described in the Materials and Methods and using the primers listed in Table 2. The new *Acanthamoeba* expression plasmids are available from Addgene.org (Table 1).

**TABLE 1** Strains and plasmids used and constructed in this study

| Strain/plasmid name | Key strain/plasmid features | Reference/source |
|---|---|---|
| DH5α | Standard *Escherichia coli* cloning strain. F⁻ *endA1 glnV44 thi-1 recA1 relA1 gyrA96 deoR nupG purB20* φ80d*lacZ*ΔM15 Δ(*lacZYA-argF*)U169, hsdR17($r_K^-$ $m_K^+$), λ⁻ | Laboratory strain collection |
| Neff | *Acanthamoeba castellanii* strain Neff. Environmental isolate from USA | ATCC 30010 |
| Neff:mEGFP-C1 | *Acanthamoeba castellanii* Neff derivative constitutively expressing mEGFP after transfection with pAcaGAPDH_mEGFP-C1 | This work |
| Neff:mEGFP-rab7a | *Acanthamoeba castellanii* Neff derivative constitutively expressing a mEGFP-Rab7A fusion protein after transfection with pAcaGAPDH_mEGFP-rab7a | This work |
| mEGFP-C1 | Mammalian expression vector, *npt*II gene expressed from SV40 and AmpR promoters; mEGFP under control of the CMV promoter and enhancer; 4,731 bp. | AddGene #54759 |
| mEGFP-N1 | Mammalian expression vector, *npt*II gene expressed from SV40 and AmpR promoters; mEGFP under control of the CMV promoter and enhancer; 4,733 bp. | AddGene #54767 |
| mCherry2-C1 | Mammalian expression vector, *npt*II gene expressed from SV40 and AmpR promoters; mCherry2 under control of the CMV promoter and enhancer; 4,750 bp. | AddGene #54563 |
| mCherry2-N1 | Mammalian expression vector, *npt*II gene expressed from SV40 and AmpR promoters; mCherry2 under control of the CMV promoter and enhancer; 4,750 bp. | AddGene #54517 |
| pAcaPrTPB_bkbn_mEGFP-C1 | mEGFP-C1 derivative with expression of the *npt*II gene under control of the *A. castellanii* TBP promoter region (193 bp); 4,588 bp. | This work |
| pAcaPrTPB_bkbn_mEGFP-N1 | mEGFP-N1 derivative with expression of the *npt*II gene under control of the *A. castellanii* TBP promoter region (193 bp); 4,590 bp. | This work |
| pAcaPrTPB_bkbn_mCherry2-C1 | mCherry2-C1 derivative with expression of the *npt*II gene under control of the *A. castellanii* TBP promoter region (193 bp); 4,572 bp. | This work |
| pAcaPrTPB_bkbn_mCherry2-N1 | mCherry2-N1 derivative with expression of the *npt*II gene under control of the *A. castellanii* TBP promoter region (193 bp); 4,573 bp. | This work |
| pAcaGAPDH_mEGFP-C1 | *A. castellanii* backbone expression vector derived from mEGFP-C1 by putting the *npt*II gene under control of the *A. castellanii* TBP promoter region and the mEGFP fusion tag expressed from the amoebal GAPDH promoter region (751 bp); 4,733 bp. | This work AddGene #208175 |
| pAcaGAPDH_mEGFP-N1 | *A. castellanii* backbone expression vector derived from mEGFP-N1 by putting the *npt*II gene under control of the *A. castellanii* TBP promoter region and the mEGFP fusion tag expressed from the amoebal GAPDH promoter region (751 bp); 4,735 bp. | This work AddGene #208182 |
| pAcaGAPDH_mCherry2-C1 | *A. castellanii* backbone expression vector derived from mCherry2-C1 by putting the *npt*II gene under control of the *A. castellanii* TBP promoter region and the mCherry2 fusion tag expressed from the amoebal GAPDH promoter region (751 bp); 4,724 bp. | This work AddGene #208178 |
| pAcaGAPDH_mCherry2-N1 | *A. castellanii* backbone expression vector derived from mCherry2-N1 by putting the *npt*II gene under control of the *A. castellanii* TBP promoter region and the mCherry2 fusion tag expressed from the amoebal GAPDH promoter region (751 bp); 4725 bp. | This work AddGene #208177 |
| pAcaGAPDH_mEGFP-rab7a | pAcaGAPDH_mEGFP-C1 derivative containing the *A. castellanii* Neff *rab7a* gene ACA1_074580 directionally cloned as *Bgl*II + *Bam*HI downstream of mEGFP to generate an in-frame translational fusion product; 5,531 bp. | This work AddGene #208179 |
| pUC18T-mini-Tn7T | Empty mini-Tn7 base vector (Ap^R) with transcriptional terminator and oriT; 3,515 bp. | AddGene #64957 |
| pTNS2 | R6K-based plasmid (Ap^R) encoding the TnsABCD Tn7 transposase expression genes; 9,615 bp. | AddGene #64968 |
| pFCM1 | High copy ColE1 plasmid (Ap^R) containing a Cm^R cassette flanked by FRT sites for FLP-mediated excision; 3,852 bp. | AddGene #64948 |
| pUC18T_mTn7TC1 | mini-Tn7 delivery vector (Ap^R) with transcriptional terminator and oriT (3515 bp). Contains the Cm^R cassette from pFMC1 cloned as a *Swa*I-*Pac*I fragment within the transposable fragment; 4,845 bp. | This work |
| pMRE-Tn7-145 | miniTn7 plasmid to deliver constitutively expressed mScarlet-I in bacteria from the class-1 integron Pc promoter. Used as the source for mScarlet-I and the Pc region. Amp^R, Gm^R, and Cm^R. Contains the tnsABCD Tn7 transposase genes; 15,443 pb. | AddGene #118561 |

**TABLE 1** Strains and plasmids used and constructed in this study (*Continued*)

| Strain/plasmid name | Key strain/plasmid features | Reference/source |
|---|---|---|
| pUC18T_mTn7TC1_Pr_mScarlet-I | Mini-Tn7 delivery plasmid to generate transcriptional fusions to mScarlet-I. The mScarlet-I gene is preceded by a RBS. The plasmid contains a chloramphenicol acetyl transferase gene encoding for chloramphenicol resistance. | This work |
| pUC18T_mTn7TC1_Pc_mScarlet-I | pUC18T_mTn7TC1_Pr_mScarlet-I derivative for chromosomal labeling of bacteria with constitutive mScarlet-I expression driven from the strong Pc promoter; 5,893 bp. | This work AddGene #208180 |
| Sm18::mTn7TC1_Pc_mScarlet-I | *S. maltophilia* Sm18 derivative tagged with mScarlet-I expressed from the strong constitutive Pc promoter. | This work |

## Phylogenomic and structural sequence analyses of 11 Rab7/Rab9 homologs in the *A. castellanii* RefSeq proteome uncovers eight (73%) non-canonical sequences

After constructing new *Acanthamoeba* expression plasmid backbones, we generated a Neff line overexpressing an mEGFP-Rab7 fusion protein to study its role in macropinocytosis of fluid phase markers and phagocytosis of live *S. maltophilia* cells. A phmmer search (see Materials and Methods) for Rab7 homologs in the reference (RefSeq) *A. castellanii* proteome (GCF_000313135.1) retrieved 15 Rab homologs (E-value ≤1e-35). We combined the phmmer hits with our curated Rab_SPhyd94 set of *bona fide* Rab reference proteins retrieved from Swiss-Prot (see Materials and Methods). The resulting data set was subjected to structure-guided multiple sequence alignment using mafft-dash, followed by maximum-likelihood (ML) phylogenetic analysis (see Materials and Methods). Eleven *A. castellanii* phmmer hits clustered in a well-supported [80% bootstrap proportion (BP)] clade delimited by the reference proteins RAB7B_HUMAN and RAB7A_HUMAN. Their sizes ranged from 184 to 211 amino acids and were selected as Rab7 homologs for further analysis. Figure 1 presents an ML phylogeny depicting the evolutionary relationships between the 11 *A. castellanii* sequences (labeled with ACA_XP_*; RefSeq codes) together with a subset of the most closely related reference Rabs from the Rab_SPhyd94 set (Swiss-Prot codes). Key features of the sequences, such as the number of C-terminal cysteines and raw FIMO scores (71) of the five Rab family specific (RabF) motifs (72, 73) (see Materials and Methods) were mapped on the tree (Fig. 1). The 11 *Acanthamoeba* RefSeq sequences had at least two significant hits ($P < 5e-04$) for RabF motifs with raw FIMO scores ≥12 (Fig. 1), as expected for *bona fide* small Rab GTPases according to the criteria defined by Surkont et al. (73). Unsurprisingly, the two Ran GTPase proteins used as outgroup sequences for tree rooting did not satisfy this criterion (Fig. 1). However, this analysis identified multiple non-canonical *Acanthamoeba* Rabs, which either lack cysteines among their five C-terminal residues (XP_004338854.1

**TABLE 2** PCR primers used in this work

| Primer name | Primer sequence (5′ to 3′) | Restriction sites | Reference |
|---|---|---|---|
| neo-Kozak_SpeI.F | aaaaACTAGTccaccatgattgaacaagatggattgcac | SpeI | This work |
| pBABE_PacI.R | aTTAATTAAaccctaactgacacacattcc | PacI | This work |
| TBP-110_PacI.F | aTTAATTAAgaaacgacgccttgcaacaagc | PacI | This work |
| TBP + 68_SpeI.R | aaaaACTAGTcttgttgtatgtgtgaatcgactcc | SpeI | This work |
| Aca_Pr_GAPDH.F2 | aaaaaATTAATgcaaccctgtgacgagagacag | AseI | This work |
| Aca_Pr_GAPDH.R | aaaGCTAGCtcttagtgagagtggtgtttgctgc | NheI | This work |
| M13pUC_PacI.F | aaaaaTTAATTAAcccagtcacgacgttgtaaaacg | PacI | This work |
| M13pUC_SwaI.R | aaaaaATTTAAATagcggataacaatttcacacagg | SwaI | This work |
| mTn7_SPPfw | aaaaATTTAAATagaCATATGtcaTTAATTAAccggacgatatcatgcatgag | SwaI, NdeI, PacI | This work |
| mTn7_SphI.Rev | aaaaATTTAAATaaGCATGCcgggccgcaagctcctag | SwaI, SphI | This work |
| ACA_rab7a.F | aaaaAGATCTatgtccacgcgaaagaaggttctg | BglII | This work |
| ACA_rab7a.R2 | aaaaGGATCCacaacgtttagcacgagcaaccc | BamHI | This work |
| SmaI_glmS_down_1549F | GACATGCCGGTGGTGGTGATCG | None | This work |
| pTn7R | CACAGCATAACTGGACTGATTTC | None | (70) |

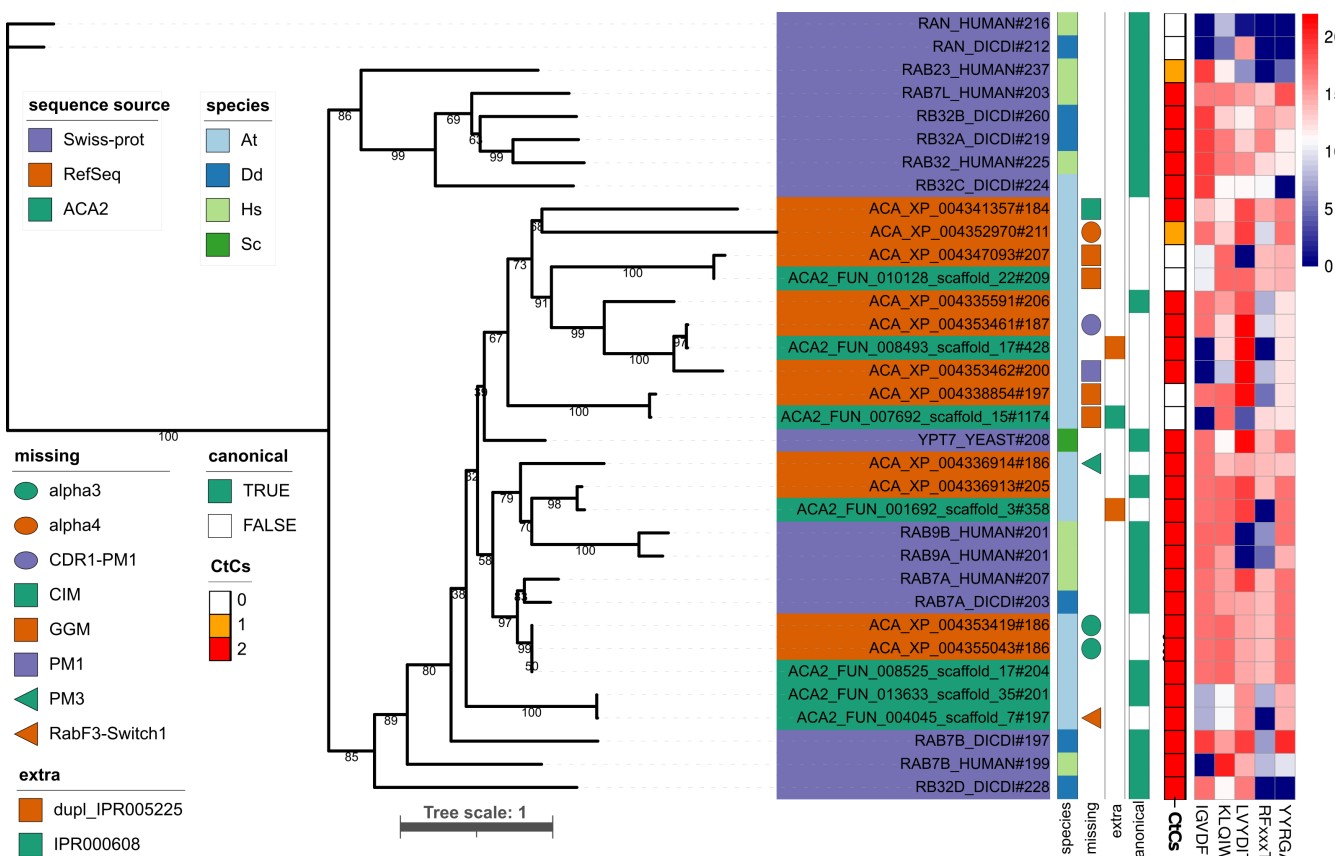

**FIG 1** Maximum-likelihood phylogeny of *A. castellanii* strain Neff Rab7 homologs found in the RefSeq and Neff-2 assemblies, along with closely related reference Rabs from humans, yeast, and *Dictyostelium discoideum* retrieved from Swiss-Prot. Structure-guided multiple sequence alignment was generated using mafft-dash and contained 34 sequences with 1,416 columns, 492 distinct patterns, 225 parsimony-informative, 170 singleton, and 1021 constant sites. ModelFinder best-fit model: LG + G4, chosen according to Bayesian information criterion. Proportion of invariable sites: 0.046; gamma shape alpha: 1.007. Tree score: −13,879.353. Bootstrap proportions for 1,000 replicates are presented below the branches. Total tree length: 23.717. Three types of sequence annotations are provided (left to right), which include (i) sequence source and species (At = *A. castellanii*; Dd = *D. discoideum*; Hs = *H. sapiens*; Sc = *S. cerevisiae*), including number of amino acids; (ii) presence-absence of conserved sequence motifs detailed in Fig. 2, presence of additional motifs and Rab classification as canonical or non-canonical sequences; (iii) FIMO scores of RabF motifs (see text for further details).

and XP_004347093.1) required as prenylation sites (72, 73) or have deletions spanning critical sequence motifs or secondary structural elements (72, 74–76). These deletions were unexpected in the highly conserved globular G-domain involved in the nucleotide-dependent structural switch characteristic of the Ras superfamily proteins (77). Figure 2 shows the structure-guided alignment of the nine *Acanthamoeba* Rab7-like homologs containing C-terminal prenylation motifs, excluding sequences with duplicated or extra domains, as shown in Fig. 1. We used ESPrit3 (78) to map secondary structural elements from the human Rab7A crystal structure (PDB acc. 1T91) to the alignment and manually added critical functional sequence motifs (Fig. 2). This analysis revealed that the *Acanthamoeba* RefSeq sequences XP_004353461.1 and XP_004353462.1 lacked the conserved nucleotide-binding motif PM1, whereas XP_004336914.1 lacked PM3 (Fig. 2A). Therefore, these sequences most likely encode non-functional small GTPases.

The two identical proteins XP_004353419.1 and XP_004355043.1, formed a tight and strongly supported clade (97% BPs) with RAB7A_HUMAN and RAB7A_DICDI (Fig. 1). In addition, these proteins were identified as reciprocal best hits for human RAB7A using the HMMSeq2 easy-rbh module (see Materials and Methods). Therefore, we conclude that these are the *A. castellanii* Rab7A orthologs. However, these sequences and XP_004336914.1 share a deletion spanning the α2 and α3 helices, which overlaps

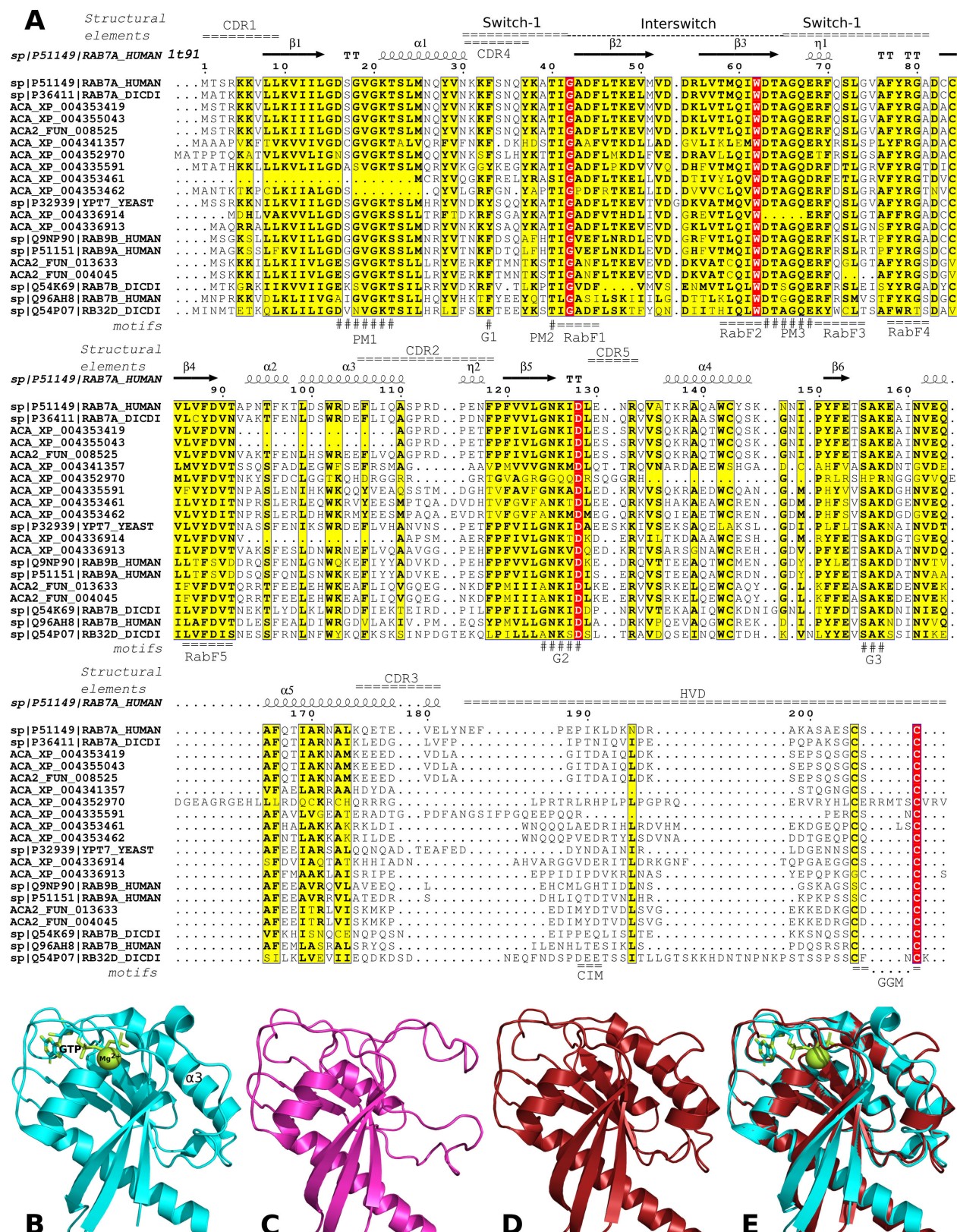

**FIG 2** Structural analysis of Rab7 homologs. (A) Structure-guided multiple sequence alignment of reference Rab7 sequences (SwissProt accessions) and *Acanthamoeba castellanii* Neff homologs from RefSeq (ACA_XP) and the Neff-2 assembly recently reported by Matthey-Doret et al. (79). Secondary structural elements of the RAB7A_HUMAN protein structure 1t91 were mapped on top of the sequence blocks using ESPrit3 (78), along with other conserved elements (Continued on next page)

**FIG 2** (Continued)

found in Rab7 proteins, including the Switch-1, Interswitch, and Switch-2 regions; complementarity-determining regions (CDR1-CDR5); and the C-terminal hypervariable domain (HVD) (76). Selected conserved sequence motifs identified in previous studies are highlighted below the sequence blocks, including the Rab family motifs (RabF1-RabF5), conserved nucleotide-binding motifs PM1-PM3 (phosphate, magnesium binding motifs), and guanine-binding motifs (G1–G3). The C-terminal interacting motif (CIM) was also annotated, along with the C-terminal cysteines involved in Rab prenylation (GGM). (B) Crystal structure of the human small GTPase Rab7 with bound GTP and $Mg^{2+}$ ions (chain 1; 1t91) retrieved from the Protein Data Bank. The α3 helix was annotated. (C) AlphaFold model (AF-L8HHF3-F1) of the *A. castellanii* Rab7A protein encoded by ACA1_074580. Notably, it lacks the α3 helix. (D) AlphaFold2 model generated in this work for the ACA2_FUN_008525 protein, which is identical in sequence to our corrected conceptual translation product for the ACA1_074580 gene with two introns. Note the presence of the α3 helix. (E) Alignment of the structures shown in B and D. All structures and structural alignments were rendered using PyMOL.

with the conserved CDR2 region (Fig. 2A through C). In addition, XP_004352970 has a deletion that fully spans the α4 helix. The AlphaFold2 model (AF-L8HHF3-F1) for the *A. castellanii* Rab7A protein (XP_004353419.1) presented in Fig. 2C revealed a distorted structure due to the lack of α2-α3 helices compared to the human 1t91 crystal structure (Fig. 2B). A global alignment of the reference 1t91 structure and the AF-L8HHF3-F1 model generated with TM-align resulted in 152 aligned residues with 75.7% identity, a root mean square deviation (RMSD) of 2.40 Å, and a template modeling score (TM-score) of 0.71339 (normalized by the length of 1t91), confirming that the AF-L8HHF3-F1 model overlaps poorly with the canonical RAB7A_HUMAN structure. Finally, protein XP_004336913.1 clustered with moderate support (82%) in the clade containing the two human Rab9 alleles, but easy-rbh did not identify the former as the reciprocal-best hit of any of the two human Rab9 isoforms (Fig. 1). In summary, 8 of the 11 phmmer hits (73%) found in the *A. castellanii* Neff RefSeq proteome are non-canonical sequences that probably encode non-functional Rab7/Rab9 homologs.

We extended our analysis to a new complete genome assembly (Neff-2) of strain ATCC 30010, recently published by Matthey-Doret et al. (79) (FUN_* labels). Only seven Rab7 homologs were found in the Neff-2 proteome using the same phmmer search parameters. FUN_00128_scaffold22 and FUN_007692_scaffold15 are devoid of C-terminal prenylation motifs and are very similar, although not identical, to XP_004347093.1 and XP_004338854.1, respectively (Fig. 1). The remaining five sequences contained at least three high-scoring FIMO hits for rabifier2 (73) RabF motifs (Fig. 1). However, Neff-2_FUN_008493 and Neff-2_FUN_001692 contained a duplication of the Small GTP-binding protein domain (IPR005225), as revealed by interproscan analysis and AlphaFold2 modeling (data not shown); therefore, they are non-canonical Rab7 homologs. The Neff-2_FUN_008525 product is a *bona fide A. castellanii* Rab7A ortholog (Fig. 1 and 2A). Finally, Neff-2_FUN_013633 and Neff-2_FUN_004045 were identical (Fig. 1), except for a three-amino acid deletion in the latter sequence, located in the unstructured region between the β3 and β4 sheets (Fig. 2A). These sequences formed an independent branch, with no counterparts in the RefSeq assembly (Fig. 1). As summarized in Fig. 1, only four non-redundant sequences out of the 16 Rab7-like proteins analyzed in this study (11 from ACA1 and 5 from Neff-2) corresponded to canonical Rab7/Rab9 homologs.

## Alignment of the RAB7A_HUMAN protein to the *A. castellanii* ACA1_074580 gene and structural modeling of the corrected conceptual product confirmed that it encodes the amoebal Rab7A ortholog

Given the universal presence of Rab7A homologs in eukaryotic cells and their highly conserved primary sequence in the G-domain (76), we hypothesized that the non-canonical and identical *Acanthamoeba* Rab7A protein sequences XP_0043533419.1 and XP_004355043.1 (Fig. 2) most likely represent sequencing or annotation errors. To discern these possibilities, we amplified and sequenced ACA1_074580 using genomic DNA purified from *A. castellanii* Neff ATCC 30010. The resulting sequence was identical to that of ACA1_074580, ruling out any sequencing errors. Miniprot (80) mapping of the RAB7A_HUMAN protein (207 aa) to the ACA1_074580 gene sequence (836 bp) suggests that it encodes three exons at the following gene coordinates 1–193, 332–677, 761–833,

followed by the TAA stop codon. The corresponding conceptual translation product encodes a 22.8-kDa polypeptide with 204 residues without the deletion affecting the α3 helix (Fig. 2C) in XP_0043533419.1 (186 aa), which is identical to Neff-2_FUN_008525. We subjected the corrected protein sequence to AlphaFold2 structural modeling (see Materials and Methods). The resulting model confirmed that it represents a *bona fide* Rab7 protein with all motifs and secondary structural elements in place (Fig. 1, 2A, D, and E).

A TM-align structural alignment of the human Rab7A crystal structure 1t91 with the AlphaFold2 model for our corrected translational product resulted in 176 aligned residues with notably lower RMSD (1.45) and higher TM-score (0.95003) than the corresponding values of AF-L8HHF3-F1 for XP_004353419.1 (RMSD = 2.40; TM-score = 0.71339; both scores normalized by the length of 1t91). The tight structural overlap revealed by the former scores is graphically confirmed in Fig. 2E, which shows the superimposed structural alignment of the corrected gene product on 1t91.

The problematic ACA1_074580 gene annotation from RefSeq contains three introns, the second being 54 nt long (at positions 412–466), which overlaps exon 2 of the corrected CDS in-frame, explaining the 18-aa deletion found in the RefSeq sequence XP_0043533419.1. Based on this evidence, we conclude that ACA1_074580 encodes a canonical Rab7A protein.

## Construction of an *A. castellanii* Neff line overexpressing the mEGFP-Rab7A fusion protein

As detailed in the Materials and Methods section, we cloned ACA1_074580 (encoding Rab7A) downstream of mEGFP into the plasmid pAcaGAPDH_mEGFP-C1 to generate pAcaGAPDH_mEGFP-rab7a (Table 1). The genetic maps are presented in Fig. 3A and B, respectively. The latter plasmid encodes the mEGFP-Rab7A fusion protein expressed from the *A. castellanii* GAPDH promoter (see Materials and Methods). The 3D structure of the conceptual translation of the fusion protein was modeled using AlphaFold2. Figure 3C shows the mEGFP and Rab7A domains separated by a short and flexible SGLRS linker peptide.

We transfected *A. castellanii* Neff with these plasmids using SuperFect and a stable transfection protocol for adherent cells (see Materials and Methods). The transfection efficiency was low, between 5% and 10%, as estimated by flow cytometry (data not shown), and the fluorescence intensity of individual trophozoites was variable. LCI confirmed the cytometry findings and revealed that, as expected, the line transfected with pAcaGAPDH_mEGFP-C1 displayed homogeneous green fluorescence throughout the cytoplasm (Fig. 3D and E) and nuclear compartments, but not in the vacuolar lumina. In contrast, strain Neff:mEGFP-rab7a (Table 1) displayed strong fluorescence localized on the surface of vesicles of varying sizes, but not in the nucleus or its membrane (Fig. 3F and G), suggesting that the fusion protein localizes to endocytic vesicles. The Neff:mEGFP-rab7a line was sorted using automated fluorescence-activated cell sorting (FACS) (see Materials and Methods) to enrich the fluorescent population (data not shown) for downstream analysis.

## Rab7A is associated with acidified late endosomes in *A. castellanii* trophozoites

Rab7A proteins are associated with acidified late endocytic vesicles including phagosomes and lysosomes (43, 81). To validate the mEGFP-Rab7A fusion (Fig. 3F and G), we used widefield fluorescence microscopy to observe Neff:mEGFP-rab7A trophozoites co-incubated with the fluorescent fluid-phase endocytic tracer dextran 10,000 (rhodamine conjugate). LCI was performed between 60 and 75 min after adding the marker, revealing that it accumulated in perinuclear Rab7A-decorated vesicles (Fig. 4A through D). This result is consistent with their classification as late endocytic vesicles.

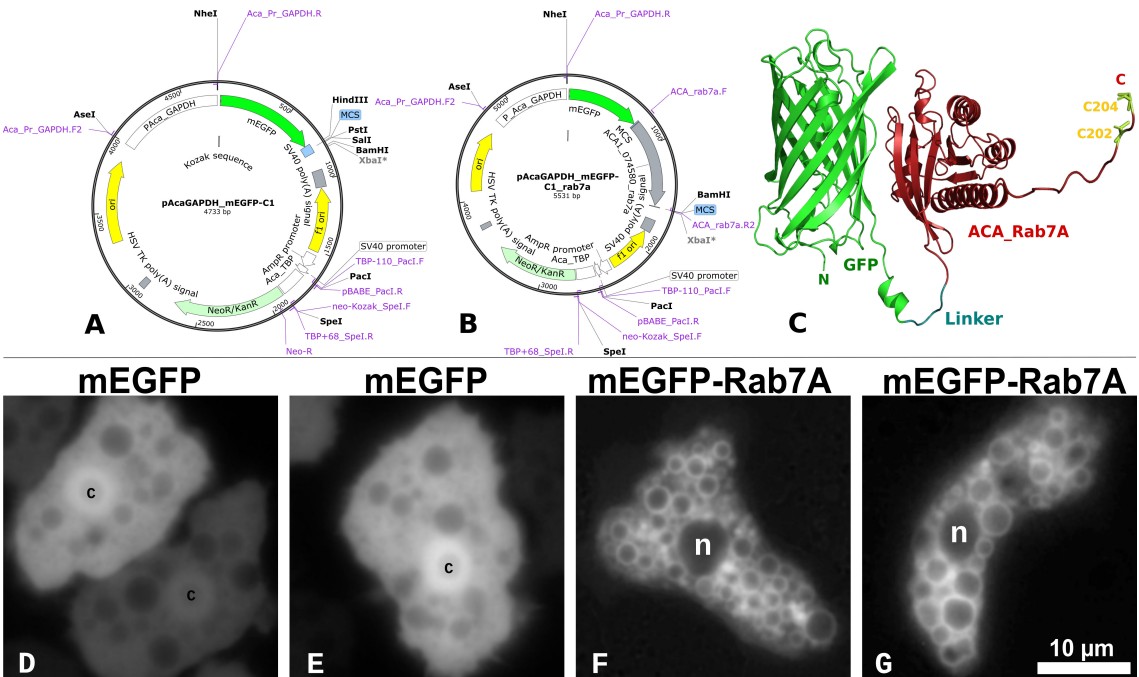

**FIG 3** Genetic maps of two *Acanthamoeba* expression plasmids constructed in this study, a predictive model of the fusion protein mEGFP-Rab7A encoded by pAcaGAPDH_mEGFP-C1_rab7A, and widefield fluorescence microscopy of live cell lines transfected with these plasmids. (A) The pAcaGAPDH_mEGFP-C1 plasmid was used to construct mEGFP-*Acanthamoeba* protein fusions expressed from the GAPDH promoter. (B) The map for pAcaGAPDH_mEGFP-C1_rab7A, with the *A. castellanii* ACA1_074580 gene fused in-frame downstream of mEGFP. Key features, primers (Table 2), and restriction sites used to construct the plasmids are highlighted on the maps. (C) AlphaFold2 model of the resulting mEGFP-Rab7A fusion protein, displaying 239 amino acids of GFP in green, followed by the SGLRS linker peptide in deep teal, and 204 residues of the corrected ACA1_074580 conceptual translation with three exons (see text) in dark red. N- and C-termini are indicated by N and C, respectively, with Rab7A C202 and C204 residues that serve as prenylation motifs displayed as yellow sticks. (D and E) *A. castellanii* Neff trophozoites expressing mEGFP at various intensities (D) from pAcaGAPDH_mEGFP-C1 display diffuse cytoplasmic and nuclear fluorescence. Nucleoli at the center of the strongly fluorescent nuclei were labeled with c. (F and G) Trophozoites expressing mEGFP-Rab7a display green fluorescence associated with multiple vesicles but not the nuclear envelope. The bar scale is the same for figures D–G, with nuclei labeled as n. All images correspond to a single optical section taken with an EC Plan-Neofluar 63×/1.25 oil M27 objective and filter set FS38 for GFP. Trophozoites were immobilized using a GelRite plug. n, nuclei; c, nucleoli.

Canonical late phagocytic vesicles are acidic, Rab7A-positive compartments (82). This was demonstrated by widefield fluorescence LCI of Neff:mEGFP-rab7A trophozoites co-incubated with pH-sensitive Zymosan–pHrodo Deep Red BioParticle conjugates (100 µg/mL) for 60 min (Fig. 4E through H). After phagocytosis, the deep-red fluorescence of the Zymosan-pHrodo BioParticles (pKa ≈5) increased as the lumen of the phagosome acidified and was negligible at neutral or alkaline pH. Consequently, the BioParticles did not fluoresce until they entered the late endosome and lysosome, as demonstrated by comparing Fig. 4E (deep-red channel, FS50) and 4H (bright field). Large non-fluorescent extracellular yeast cell wall (Zymosan) BioParticles are highlighted in the latter by black arrows. Figure 4G depicts the merged deep-red and GFP channels, demonstrating that intracellular bright fluorescent Zymosan-pHrodo particles were enclosed in Rab7A-positive vacuoles.

From these experiments, we concluded that the mEGFP-Rab7A fusion protein correctly labeled acidified, late endocytic vacuoles, possibly including phagolysosomes, and that *A. castellanii* strain Neff could effectively engulf yeast-sized cells with a median cell volume of 42 µm$^3$, which is approximately 32 times larger than the estimated volume of 1.3 µm$^3$ for an *E. coli* cell (83).

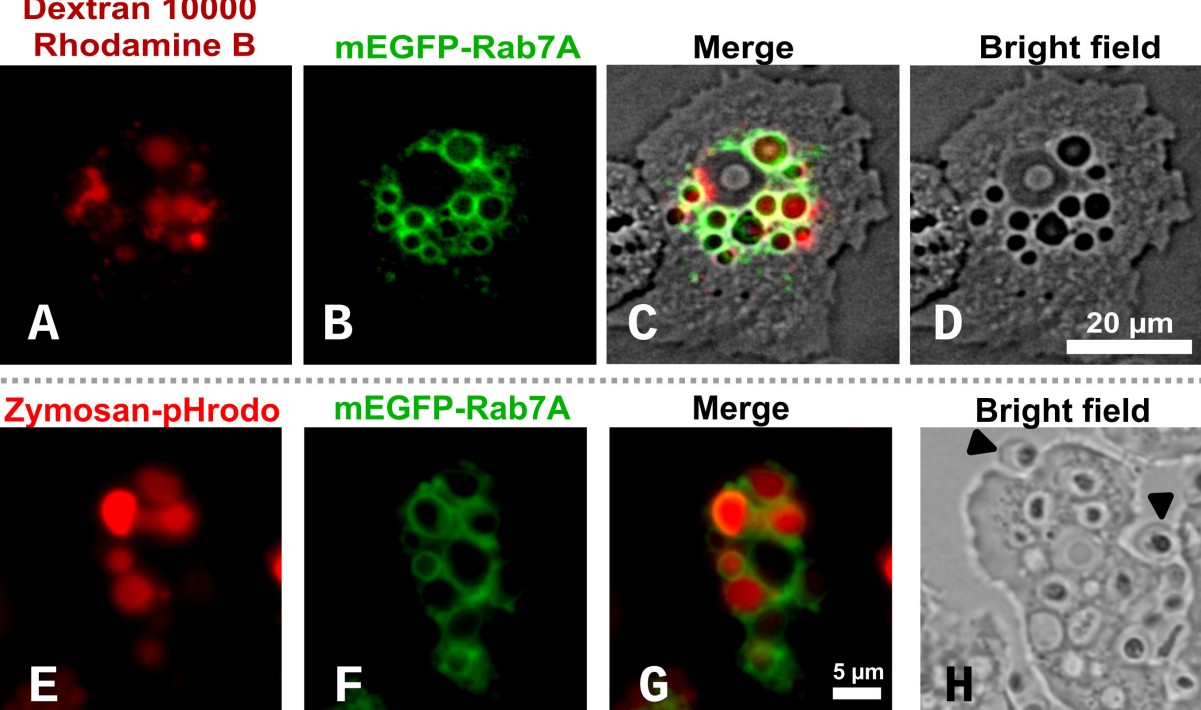

**FIG 4** Widefield fluorescence microscopy of Neff:mEGFP-Rab7A trophozoites demonstrating that endocytic (A–D) and phagocytic (E–H) vesicles are delimited by Rab7A-positive membranes. (A) Red fluorescence of the fluid-phase endocytic tracer dextran 10,000-rhodamine B. (B) Green channel, revealing fluorescence of the mEGFP-Rab7A fusion associated with peri-nuclear vesicles. (C) Merged red, green, and bright field channels, demonstrating the colocalization of dextran particles within Rab7A-positive vacuoles. (D) Bright field image. Live cells growing in six-well polystyrene dishes were imaged (A–D) as a single optical section with a long-distance LD Plan-Neofluar 63×/0.75 Corr Ph2 M27 objective, and trophozoites immobilized with a GelRite plug. (E) Deep-red fluorescence of pH-sensitive Zymosan-pHrodo BioParticles used as phagocytic tracers that increase their fluorescence intensity with decreasing pH. (F) Green fluorescence emitted by the Rab7A-positive vacuoles. (G) Merged deep-red and GFP channels, demonstrating that acidic Zymosan-pHrodo-containing vesicles are enclosed within Rab7A-positive membranes. (H) Bright field image with black arrowheads pointing to extracellular Zymosan particles that do not fluoresce in neutral culture medium (pH 7.2). LCI for (E–H) was performed in glass-bottom dishes using an EC Plan-Neofluar 63×/1.25 oil M27 objective and filter sets listed in the Materials and Methods section. The scale was the same for all four images. Trophozoites were immobilized with a GelRite plug.

### *Acanthamoeba castellanii* Neff trophozoites host *Stenotrophomonas maltophilia* Sm18 cells within acidified Rab7A-positive phagosomes

Our next goal was to investigate the fundamental cell biological aspects of the *A. castellanii-S. maltophilia* interactions. Given that intracellular pathogens deploy various strategies to subvert the canonical phagocytic pathway to evade digestion in phagolysosomes (18), we aimed to determine whether *S. maltophilia*-containing vacuoles (SmCVs) are Rab7A positive and acidified. To answer these questions, we established co-cultures of Neff:mEGFP-rab7A trophozoites with live *S. maltophilia* Sm18 (wild type) cells at a multiplicity of infection (MOI 50) stained with pH-sensitive pHrodo Red succinimidyl ester (Sm18-pHrodo) dye (pKa = 6.5). We performed widefield fluorescence LCI at 1 and 3 h post primary contact (ppc). Figure 5A depicts representative trophozoites showing that Rab7A-positive vacuoles of varying sizes delimit red-fluorescent SmCVs. Their variable fluorescence intensities suggest that the corresponding vacuoles may differ at (acidic) luminal pH. However, some of this variation may result from vacuoles located in slightly different focal planes. To minimize this effect, three optical sections were imaged. These experiments provided strong and direct evidence that *S. maltophilia* cells are enclosed in acidic Rab7A-positive SmCVs. At 3 h ppc, no extracellular bacteria were observed, implying that most of the Sm18-pHrodo cells were phagocytosed, explaining the higher proportion of red-fluorescent vacuoles found at this later time.

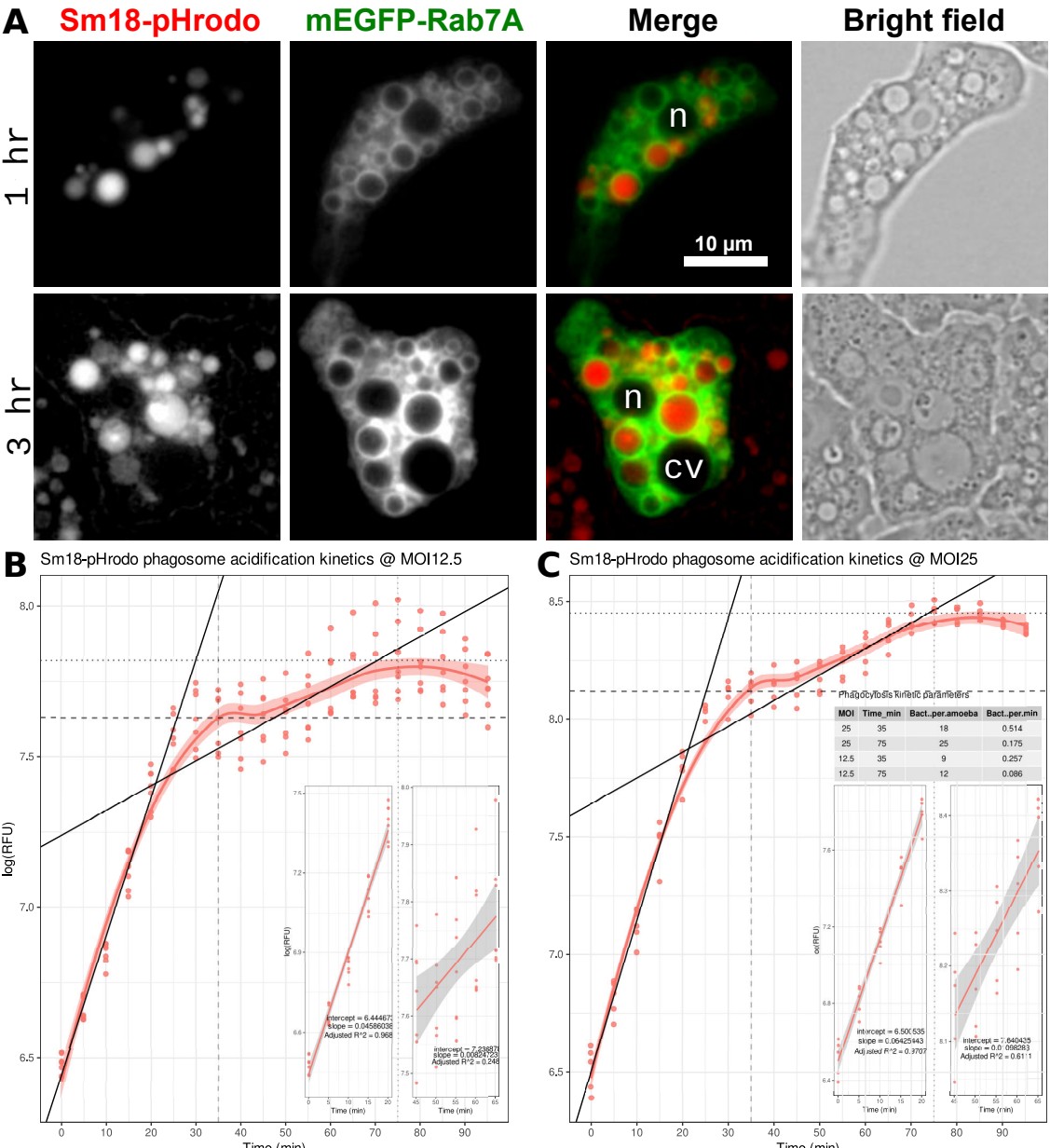

**FIG 5** LCI and statistical analysis of *A. castellanii*:mEGFP-Rab7A phagosome acidification kinetics hosting live *S. maltophilia* Sm18 cells stained with pHrodo red succinimidyl ester at different MOIs. (A) LCI of cocultures at 1 and 3 h post primary contact (ppc) (MOI 50). The two panels depict representative trophozoites imaged in red (FS 43, pHrodo Red), green (FS 38, mEGFP-Rab7A), and bright field channels. pHrodo Red signals with variable intensities were clearly visible, indicating that Sm18 was enclosed in the acidic compartment as early as 1 h ppc. The merged red and green channels (third column) demonstrate that the red-fluorescent acidic vacuoles were delimited by Rab7A-positive membranes. LCI was performed in glass-bottom dishes using an EC Plan-Neofluar 63×/1.25 oil M27 objective lens. The scale is the same for all images, each representing a composite of three optical sections stacked with the manual extended depth-of-focus Zen module. Trophozoites were partially immobilized with a GelRite pad. (B and C) Log-transformed pHrodo Red readouts of the co-cultures collected every 5 min in microplate assays used to measure phagosome acidification kinetics at MOI 12.5 (B) and MOI 25 (C). Both curves revealed a bacterial concentration-dependent response of the log-transformed red fluorescence readout. The "S-shaped curve with two inflection points" or "double-sigmoid curve" was fitted with a loess ("locally weighted scatterplot smoothing", a non-parametric method of fitting a curve to the data using a sliding window to estimate the local polynomial regression) smoother function. The insets show linear models fitted to the two linear (log-transformed) segments of the curve preceding the first and second relative fluorescence units (RFU) inflection points. The corresponding regression lines are plotted on a double-sigmoid curve to show the acidification rates (slopes) of the two segments. The tabular inset in C summarizes the kinetic parameters estimated from the two average fluorescence maxima at 40 and 80 min, as indicated by the horizontal dashed and dotted lines, respectively. n, nuclei; cv, contractile vacuoles.

## Microtiter plate co-culture assays with live Sm18-pHrodo red cells to measure phagosome acidification kinetics

To gain insight into the kinetics of phagosome acidification, we established a micro-plate assay to follow the fluorescence readout of live *S. maltophilia* cells stained with pHrodo Red succinimidyl ester (Sm18-pHrodo) hosted in *Acanthamoeba* phagosomes. As controls, we included wells containing only the labeled bacteria. Assays were performed in 96-well plates with static incubation at 30°C, using a multimodal plate reader. Eight wells per treatment were seeded with 1e + 05 trophozoites in 100 µL of MMsalts-MOPS-Glc and incubated for 30 min at 30°C to allow trophozoite adhesion. The medium was then replaced with a suspension of Sm18L-pHrodo cells in 100 µL of MMsalts-MOPS-Glc at different MOIs (12.5, 25, and 50), followed by centrifugation to synchronize phagocytosis and placement in a plate reader. Phagosome acidification was measured following the red fluorescence readout every 5 min for 90 min. The experiments were repeated three times.

Figure 5B and C depict the log-transformed phagosome acidification curves obtained for co-cultures at MOI 12.5 and MOI 25, respectively. Their shapes resemble a double-sigmoid curve ("S shaped" with two inflection points), as suggested by the Loess (non-parametric) smoother function fitted to the data. The curves revealed two fluorescence (acidification) maxima over 90 min of the experiment. The first peak on both curves was preceded by an approximately linear segment between 0 and 20 min, with slopes of 0.0459 and 0.0643 for the MOI 12.5 ($R^2 = 0.968$) and MOI 25 ($R^2 = 0.971$) curves, respectively (inset I in Fig. 5B and C). These segments correspond to the maximum acidification rates resulting from the maximum bacterial uptake rate attained by the combined effects of the highest external bacterial density and their synchronized deposition on trophozoite cells by centrifugation. Notably, the first peak was reached at approximately 35 min on both curves (~40 min, considering the previous centrifugation step), suggesting that this is the time required by *Acanthamoeba* phagosomes to reach maximal acidification when exposed to live *S. maltophilia* cells. A second peak was reached on both curves 40 min later, followed by a decline in mean fluorescence shortly thereafter. This peak was reached after a second approximately linear segment of the log-transformed data between 45 and 65 min on both curves. Fitting linear models to the corresponding data points revealed a lower slope (acidification rate) for these segments on both curves (0.00824 and 0.00983, respectively). This slower rate likely reflects that the trophozoites required increasingly longer times to encounter the remaining extracellular bacteria at this later time interval. The second fluorescence peak was reached at approximately 80 min ppc, which probably corresponded to the maximum bacterial load in the phagosomes. Based on these assumptions and estimates, we calculated that at ~40 min ppc and the lowest MOI of 12.5, the trophozoites contained, on average, nine cells, ingested at a maximum phagocytic rate of 0.225 bacteria min$^{-1}$ trophozoite$^{-1}$. The corresponding kinetic estimates for the segment preceding the second peak are shown in the inset of Fig. 5C.

## New mini-Tn7 delivery plasmids for stable tagging of *Stenotrophomonas maltophilia* cells with mScarlet-I

We used the pUC18T_mTn7T empty backbone plasmid described by Choi et al. (70) to develop improved mini-Tn7 delivery plasmids for *S. maltophilia* encoding chloramphenicol (Cm) resistance (see Materials and Methods), including pUC18T_mTn7TC1_Pc_mScarlet-I, which constitutively expresses mScarlet-I (67) from the strong Pc promoter of class 1 integrons (68). For our environmental *Stenotrophomonas* isolates (54), including *S. maltophilia* Sm18, Cm is one of the few antibiotics useful as a selective agent. Genetic maps of the two mScarlet-I plasmids constructed in this study are shown in Fig. 6A and B. We used pUC18T_mTn7TC1_Pc_mScarlet-I (Table 1) to stably tag the chromosome of strain Sm18 (see Materials and Methods), generating strain Sm18::mTn7TC1_Pc_mScarlet-I (Table 1). The proper site and stability of the mTn7 insertion were determined as described in the Materials and Methods. Figure 6C shows three trophozoites containing

red-fluorescent Sm18::mTn7TC1_Pc_mScarlet-I cells (MOI 25) within phagocytic vacuoles of different sizes. This experiment confirmed that the fluorescent signal emitted by tagged Sm18 cells was sufficiently bright to be detected using standard widefield fluorescence microscopy at the lowest MOI evaluated.

We performed additional experiments to evaluate the potential fitness burden imposed by the mTn7 insertion on the tagged strain. Non-parametric Wilcoxon tests of two growth curve parameters (t_gen and t_mid) for the parental and tagged strains cultivated in Luria-Bertani (LB) at 30°C were not significant (Fig. 6D), suggesting that the expression of *cat* (Cm$^R$) and fluorescent protein cassettes does not impose a significant fitness burden on the mutant. We evaluated the biofilm formation capacity of Sm18 and its tagged derivative in 96-well microtiter plates at 30°C and 37°C using a standard crystal violet (CV) assay. As shown in Fig. 6E, no significant differences in biofilm formation capacity were found between the two strains or at different temperatures, as determined by the Wilcoxon test.

## *S. maltophilia* Sm18 establishes an intracellular replication niche in *A. castellanii* Neff

Experiments described in the previous sections demonstrated that *S. maltophilia* Sm18 cells are hosted within acidic Rab7A-positive SmCVs. We designed a final set of experiments to determine whether Sm18 replicates within trophozoites, based on the fluorescent readout of our Sm18::mTn7TC1_Pc_mScarlet-I strain in co-cultures with amoeba. Briefly, eight technical replicates per biological replicate (*n* = 3) of the Neff/

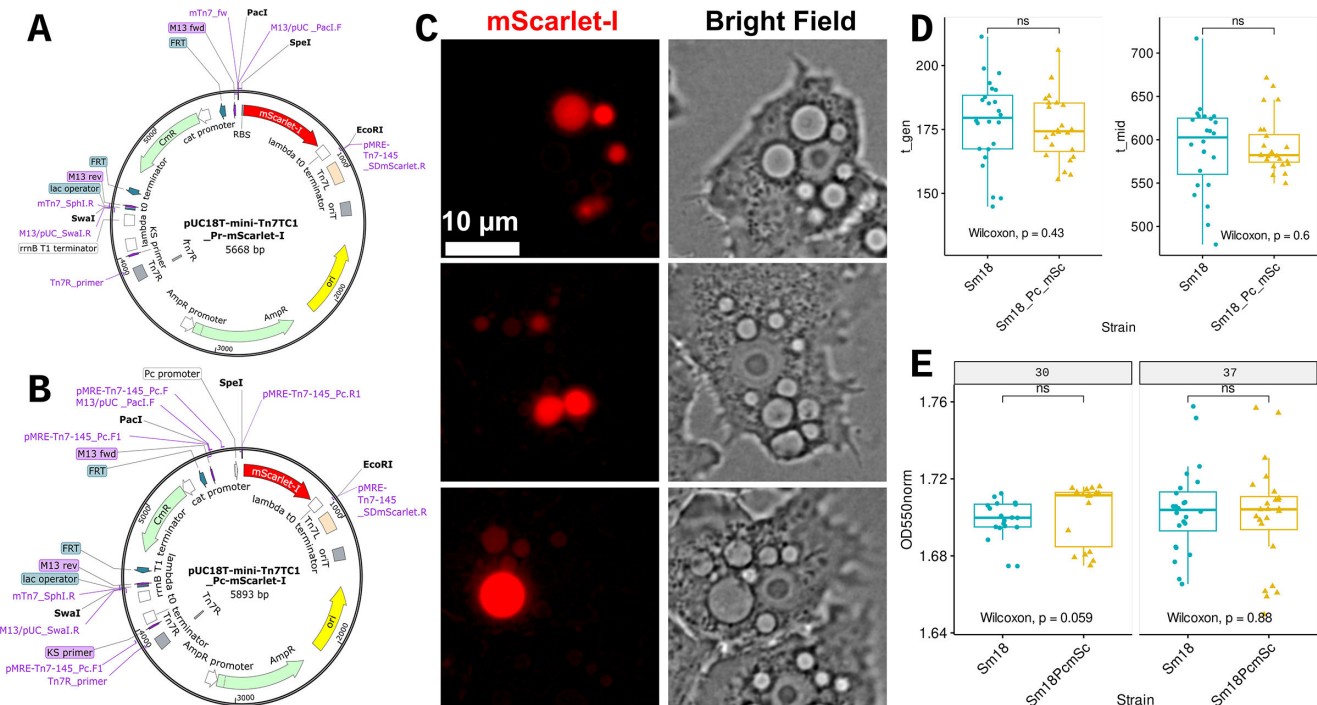

**FIG 6** Genetic maps of the two mini-Tn7 delivery plasmids constructed in this study and phenotypic analysis of the insertion mutant strain Sm18::mTn7_Pc_mScarlet-I. (A) Genetic map of the promoterless plasmid pUC18T-mTn7TC1_Pr-mScarlet-I. (B) Genetic map of pUC18T-mTn7TC1_Pc-mScarlet-I with a strong Pc promoter that drives the constitutive expression of mScarlet-I. The key features, primers, and restriction sites used to construct the plasmids are highlighted in the map. (C) LCI using widefield fluorescence microscopy of the interaction between *A. castellanii* Neff (wild type) trophozoites and Sm18::mTn7_Pc_mScarlet-I (MOI 25, 1 h ppc). LCI was performed using an EC Plan-Neofluar 63×/1.25 oil M27 objective. (D) Statistical analysis (non-parametric Kruskal-Wallis test) of t_gen (doubling time) and t_mid [time at which the population density reaches one-half of its carrying capacity (1/2 K)] growth parameters for Sm18 (wild type) and the Sm18::mTn7_Pc_mScarlet-I mutant. (E) Statistical analysis (non-parametric Wilcoxon test) of the biofilm-forming capacity of Sm18 and the mTn7 insertion mutant on polystyrene 96-well microplates measured with the crystal violet assay (OD550norm = 550 nm/620 nm) at 30°C and 37°C, 48 h pp. The growth and biofilm experiments consisted of six biological replicates with three technical replicates each.

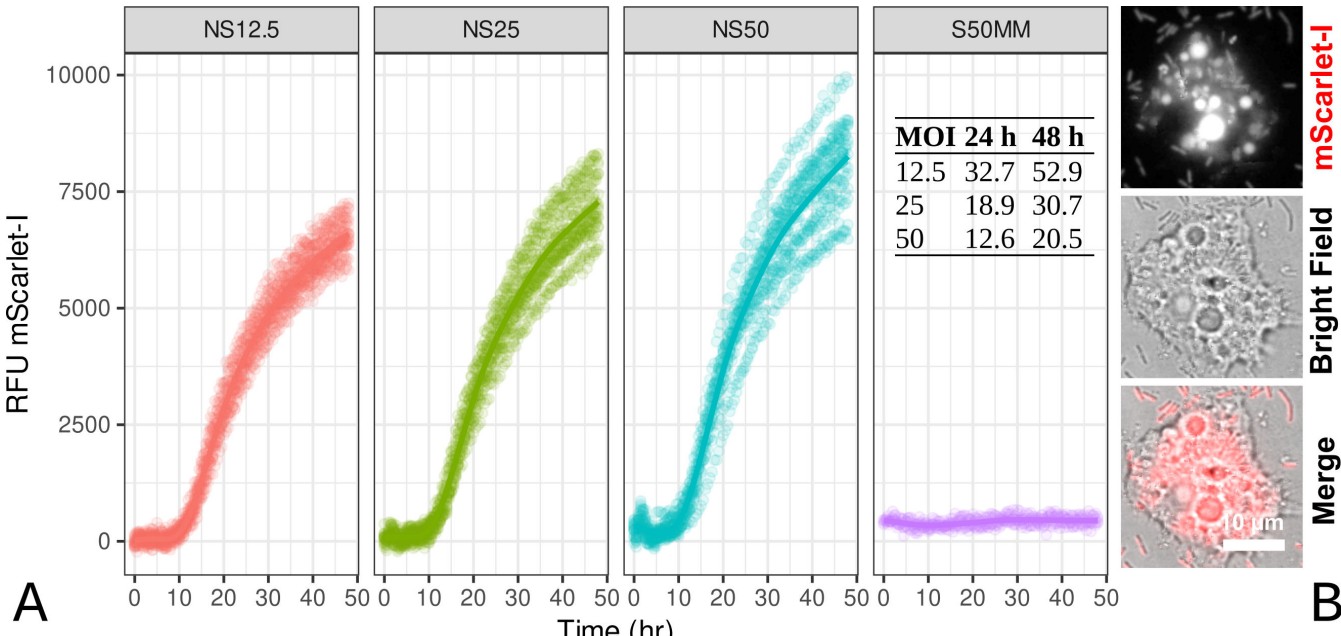

**FIG 7** Intracellular replication assays of *S. maltophilia* Sm18::mTn7TC1_Pc_mScarlet-I in co-culture with *A. castellanii* strain Neff trophozoites. (A) Co-cultures were set up at three MOIs (12.5, 25, and 50), synchronized by centrifugation, and statically incubated at 25°C in a multimodal plate reader. The fluorescence readout of mScarlet-I was measured every 30 min for 48 h. The control panel (S50 MM) revealed that Sm18::mTn7TC1_Pc_mScarlet-I could not grow in the MMsalts-MOPS-Glc medium. The tabular inset shows the median fluorescence increase factors measured at 24 and 48 h for the three MOIs. (B) Representative trophozoites imaged at 24 h ppc using the indicated channels.

Sm18::mTn7TC1_Pc_mScarlet-I cocultures were established in 96-well plates at MOI 12.5, MOI 25, MOI 50, and MOI 100 and incubated statically at 25°C (to minimize biofilm formation) for 48 h in a plate reader (see Materials and Methods). We included control wells containing only the medium, trophozoites, or Sm18::mTn7TC1_Pc_mScarlet-I cells, corresponding to the numbers of cells used as inoculum at different MOIs. Red fluorescence emitted by mScarlet-I was recorded every 30 min for 48 h. Figure 7A shows the evolution of fluorescence over time for different MOIs. At MOI1 2.5, we observed an initial lag phase of ~6 h, after which the red fluorescence increased sharply and continued to increase linearly for approximately 12 h. After this time point (~18 h), the replication (fluorescence) rate slowly and steadily decreased but did not reach an asymptote within 48 h. We estimated the number of bacterial replication cycles at 24 h and 48 h ppc for the three MOIs based on the median RFUs (see the tabular inset in Fig. 7A).

Microscopic examination of the cultures at 24 h ppc revealed active healthy trophozoites containing multiple red fluorescent vacuoles. However, abundant extracellular bacteria were also observed (Fig. 7B). Sm18::mTn7TC1_Pc_mScarlet-I cells did not grow either in the standard MMsalts-MOPS-Glc medium (S50MM panel in Fig. 7A) or in 24-h-old MMsalts-MOPS-Glc filtered (0.24-μm membrane) medium from Neff trophozoite cultures (conditioned medium, data not shown). These results suggest that *S. maltophilia* Sm18 can establish an intracellular replication niche within trophozoites at the three MOIs evaluated, eventually returning to the extracellular medium via a non-characterized non-lytic exocytic pathway.

## DISCUSSION

In this study, we developed a new set of expression plasmids for the generation of C- and N-terminal fusions of *Acanthamoeba* proteins to mEGFP (52) and mCherry2 (84), two FPs that are truly monomeric. We adapted the empty backbone mammalian plasmids from Michael Davidson's group (Table 1) by replacing the viral promoters

driving the expression of *nptII* and fluorescent proteins with *Acanthamoeba* TBP and GAPDH promoters, respectively (Fig. 3A and B). These promoters were first evaluated by Peng et al. (69) and chosen to construct pGAPDH-EGFP (48). We selected them based on empirical evidence of their usefulness in *Acanthamoeba* expression analysis of diverse fusion proteins by fluorescence microscopy, such as a cytoplasmic M17 leucine aminopeptidase critical for encystation (49), cyst wall lectins (50), and the nuclear nicotinamide adenine dinucleotide-dependent deacetylase AcSir2 protein (51), among others. A recent publication reported that the GAPDH promoter has a stable and moderately strong expression in *A. castellanii* under various conditions, as evaluated by quantitative real-time PCR (qPCR) (85). Therefore, although we did not directly test the promoter strength or quantify the amount of Rab7A-EGFP fusion protein produced, we assumed that it was overexpressed, given the strong fluorescent signal detected. Although convenient for microscopy, constitutive Rab7A overexpression in hamster and human fibroblasts has been reported to lead to the formation of larger vacuoles that tend to be more scattered throughout the cytoplasm (43, 86). Despite the enrichment of fluorescent Neff:mEGFP and Neff:mEGFP-Rab7A cells subjected to FACS, such a pattern was not consistently observed in our study, as the number, size, and position of Rab7A-positive vacuoles varied across cells. This variability may result from the high ploidy of the Neff genome (~25×) (9), which is subjected to amitotic nuclear divisions, resulting in aneuploidy and the random segregation of genetic material (79, 87). However, we observed that long-term (>18 h) co-culture of Neff:mEGFP-Rab7A with Sm18 cells resulted in the lysis of a fraction of trophozoites due to bacterial release into the cytosol, suggesting that overexpression of the mEGFP-Rab7A fusion protein destabilizes phagosomes (data not shown). As discussed later, this was not observed in wild-type Neff co-cultures with Sm18.

In contrast, TBP promoter expression has been reported to be less stable (85). In our study, the transfection efficiency was low (5%–10%). A promoter that drives expression of the *nptII* gene at higher and more stable levels would be desirable, as it might enhance the selection efficiency of the transfected lines. Interestingly, the FACS-enriched populations of fluorescent cells tolerated higher G418 concentrations probably because they contained a higher number of plasmids. However, fluorescence gradually diminished with increasing subculture passages, despite antibiotic selection.

We demonstrated the usefulness of the plasmid pAcaGAPDH_mEGFP-C1 for studying the phagocytic pathway of *A. castellanii* Neff (ATCC 30010) (45, 79) in its interaction with *S. maltophilia* ESTM1D_MKCAZ16_6a (Sm18). The latter is an environmental multidrug-resistant strain recovered from the sediments of a clean river in Morelos, central Mexico, of multilocus sequence type 139 (ST139) (54). This strain is one of the few known environmental representatives of the well-defined and perfectly supported *S. maltophilia sensu stricto* multilocus sequence analysis clade 6, which holds almost exclusively clinical isolates, including the reference strain *S. maltophilia* K279a (ST1) and type strain ATCC 13637[T] (ST14) (54, 88).

*S. maltophilia* is one of the top 10 most prevalent multidrug-resistant opportunistic nosocomial pathogens (55). This species has recently been shown to be a persistent colonizer of murine non-lymphoid tissue-resident macrophages, demonstrating that it can evade the immune response by hiding within mammalian colonic and bone marrow-derived macrophages (89). Despite its clinical importance, comparatively little is known about the virulence mechanisms displayed by *S. maltophilia* (64), including the molecular strategies it deploys for intracellular survival in diverse professional phagocytes such as FLA and macrophages. In this study, we investigated the fundamental cell biological and physiological properties of *A. castellanii* SmCVs using LCI combined with a set of microtiter plate assays to gain qualitative and kinetic insights into phagosome maturation and intracellular bacterial replication.

Given the high evolutionary conservation of the phagocytic pathway in eukaryotes (90, 91), we hypothesized that Rab7 is critical for phagosome and lysosome maturation in *A. castellanii*, as has been reported in humans and *D. discoideum* (81, 92). Phylogenomic

studies of Rab GTPases in *A. castellanii* concluded that it contains 93 Rabs (38) and that the Rab7 family has experienced significant expansion, with six paralogs uniquely found in this species (39). Our phylogenomic analysis of Rab7 family members in the reference *A. castellanii* genome (RefSeq GCF_000313135.1) identified 11 members (Fig. 1) as recently reported by Porfírio-Sousa et al. (39). However, careful inspection of these sequences revealed that eight (73%) did not comply with the known structural features required for canonical small Rab GTPases to be functional (76). Among them was the *A. castellanii* Rab7A ortholog XP_004353419.1, which lacked the entire α3 helix (Fig. 1 and 2). We hypothesized that this non-canonical sequence represents a sequencing and/or annotation error as it displays an altered Rab surface that is critical for interactions with effector proteins recruited by Rabs (76) (Fig. 2C). Alignment of the RAB7A_HUMAN protein against the *Acanthamoeba* ACA1_074580 gene using miniprot (80) revealed that the gene had three exons and two introns instead of three introns, as suggested by the RefSeq annotation. AlphaFold2 modeling (93, 94) of the miniprot-corrected conceptual translation product of ACA1_074580 demonstrated annotation errors and restoration of the missing α3 helix (Fig. 2D).

Thorough structural and phylogenomic evaluation of *A. castellanii* Rab7 paralogs allowed us to confidently conclude that ACA1_074580 encodes the RAB7A_HUMAN ortholog. The entire gene sequence was amplified and directionally cloned in frame downstream of the mEGFP gene of pAcaGAPDH_mEGFP-C1 to generate pAca-GAPDH_mEGFP-rab7a (Table 1; Fig. 3B). We used this plasmid to generate the Neff:mEGFP-Rab7A line, which stably expressed the mEGFP-Rab7A fusion protein from the GAPDH promoter (Fig. 3B and C). We showed that red-fluorescent dextran, a classic fluid-phase tracer for macropinocytosis (non-specific, receptor-independent endocytosis) (95), co-localized with Rab7A-positive macropinosomes (Fig. 4A through D). The LCI of Neff:mEGFP-rab7a fed with pH-sensitive Zymosan pHrodo Deep Red conjugate BioParticles confirmed that the mEGFP-Rab7A fusion protein decorated the acidified phagosomes (Fig. 4E through H). Together, these results demonstrated the correct localization of the mEGFP-Rab7A fusion protein to macropinocytic and phagocytic vacuoles, as reported in other eukaryotic organisms (96, 97). A limitation of our simple LCI experimental setup is that it cannot provide a precise quantitative estimate of the luminal pH of the phagosomes. However, the pKa of pHrodo Deep Red is ≈5.0, suggesting that the phagosomes with the brightest signal (Fig. 4E) most likely had a pH ≤5.0. This estimate is consistent with the early pH measurements of pinocytic and phagocytic vacuoles of *A. castellanii* Neff, which were estimated to be approximately 4.8 (98).

After validating the Neff:mEGFP-rab7a line, our next goal was to determine the essential aspects of the cellular microbiology of *Acanthamoeba-Stenotrophomonas* interactions using LCI. Based on previously published evidence from transmission electron microscopy (56, 59), we expected that *S. maltophilia* would be hosted in vacuoles. However, two classes of vacuolar pathogens exist: those that occupy vacuoles resembling canonical mature phagosomes and those that create a PCV that rapidly disconnects and diverges from the canonical endocytic pathway (99). To classify *S. maltophilia* into one of these broad groups, we performed co-culture experiments with Neff:mEGFP-rab7a and live Sm18 cells stained with the pH-sensitive dye pHrodo Red succinimidyl ester at three MOIs. LCI of the interaction at 1 and 3 h ppc unambiguously revealed that Sm18 was hosted within acidic Rab7A-decorated vacuoles (Fig. 5A). Therefore, given that Rab7A is a crucial marker of late phagosomes (42, 43, 86) and that acidification is a hallmark of phagosome maturation, a process subverted by diverse pathogens (99), we concluded that *S. maltophilia* belongs to the class of vacuolar pathogens that extensively interacts with the endocytic pathway (Fig. 5A).

A limitation of our LCI approach, which is based on manual microscopy (see Materials and Methods), is that it cannot provide kinetic estimates of bacterial uptake or phagosomal acidification. To circumvent this limitation, we developed microtiter plate co-culture assays with pHrodo-labeled bacteria using an automated plate reader. Using this experimental setup, we determined that phagosome acidification reaches its first peak

at approximately 40 min ppc. Acidification was measured as early as 5 min ppc for all MOIs tested. The maximum uptake rate of live Sm18 cells occurred within the first 20 min of the assay. At an MOI of 12.5, we estimated the bacterial uptake rate to be approximately 0.225 bacterial cells $min^{-1}$ $trophozoite^{-1}$. Our kinetic estimates were very close to those provided by early flow cytometry studies on phagocytosis of *A. castellanii* (an undefined strain) using fluorescent latex microbeads (100). They found that the total number of phagocytosed beads increased approximately linearly over the first 30 min at ~22 beads 100 $cells^{-1}$ $min^{-1}$, reaching a peak at ~30 min (100). Ours is the first work to use pH-sensitive pHrodo BioParticles and live staining of *S. maltophilia* cells with pHrodo red succinimidyl ester to analyze phagosome acidification in *Acanthamoeba* by LCI and microtiter plate assays. This combination is valuable because it provides qualitative single-cell and subcellular details and high-throughput kinetic population-level data of the interaction using widely accessible equipment.

After demonstrating that *S. maltophilia* Sm18 is hosted within rapidly acidifying phagosomes, we wanted to determine its fate after longer co-culture times with Neff trophozoites. To facilitate these studies, we constructed pUC18T_mTn7TC1_Pc_mScarlet-I, which encodes a transposable element conferring Cm resistance and constitutively expresses mScarlet-I (67) from the strong Pc promoter of class 1 integrons (68). We chose mScarlet-Ibecause it is a rapidly maturing, bright, acid-tolerant, genuine monomeric red fluorescent protein derived from mRED7, a synthetic construct (67). This plasmid was then used to generate Sm18::mTn7TC1_Pc_mScarlet-I. In line with a recent publication that developed $Gm^R$ mTn7-based delivery plasmids optimized for sfGFP, mCherry, tdTomato, and mKate2 tagging of clinical *S. maltophilia* isolates (101), we found that Tn7 insertions were stably inherited in the absence of antibiotics and did not exert a significant fitness burden on diverse phenotypes (Fig. 6). In contrast to these plasmids, which express FPs with codon usage adapted to *Xanthomonadaceae*, codon usage in pUC18T_mTn7TC1_Pc_mScarlet-I was not modified and should, therefore, be useful to tag a broad range of Proteobacteria.

Using Sm18::mTn7Pc_mScarlet-I and a microtiter plate co-culture assay, we showed that Sm18 replicates within *A. castellanii* trophozoites at different MOIs without causing amoebal lysis. The absence of Sm18 replication in control wells, including a "conditioned" version of our standard co-culture medium, strongly suggested that replication occurred intracellularly. Replication commenced following a lag phase, as previously reported in a comparable experimental setup employing *Legionella pneumophila* (102). Notably, LCI of co-cultures performed at 24 and 48 h ppc revealed an abundance of extracellular bacteria at all evaluated MOIs. Nevertheless, the trophozoites remained intact and active, indicating that the bacteria were released via a non-lytic exocytic pathway, resembling the process observed in *Vibrio cholerae* (23) or the vomocytosis pathway described for *Cryptococcus neoformans* (103, 104). Alternatively, it may involve expelled food vacuoles, as observed in diverse amoebae and ciliates that contain live bacteria within membrane-bound vesicles (105). However, the latter route is unlikely, as we were unable to detect extracellular vesicles. Further research, including time-lapse LCI, is warranted to determine the cellular and molecular mechanisms governing the exit pathway of ingested *S. maltophilia* into the extracellular environment.

Our investigation did not yield conclusive proof of replication in the contractile vacuole, as demonstrated in *V. cholerae* (23) and *Bordetella bronchiseptica* (29). Future experiments conducted in low osmotic media may provide insight into whether such conditions promote the invasion of contractile vacuoles, which were not well developed in the MMsalts-MOPS-Glc medium used herein. This medium was chosen because it contains the full complement of macro-, micro-, and trace elements of a microbiological-defined medium, including $Ca^{2+}$ and $Cl^-$ ions required for adhesion, signaling, and anion transport across membranes as well as the transition metals Cu, Fe, Mn, and Zn. These metals are manipulated by professional phagocytes, subjecting ingested bacteria to metal starvation and intoxication (106, 107), suggesting that *S. maltophilia* possesses effective mechanisms for coping with such stressors.

Our results demonstrate that SmCVs belong to a class of PCVs that partially resemble mature phagosomes that interact extensively with the endocytic pathway (99). However, our understanding of SmCV formation and maturation remains limited. Notably, the potential contributions of secretion systems and their effectors to the subversion of the late phagocytic pathway in *Acanthamoeba* remain unknown. In addition to Rab5 and Rab7A, a network of other Rabs is known to orchestrate phagosome maturation in diverse organisms, with Rab10, Rab11, Rab14, and Rab32 being the most conserved (97). However, their functions have yet to be characterized in *Acanthamoeba*. With regard to Rab7 paralogs, a recent report showed that XP_004353462.1, one of the non-canonical Rab7 homologs with a deletion in the PM1 motif (Fig. 2A), is induced in *A. castellanii* trophozoites fed with *E. coli* DH5α (40). Interestingly, Kim et al. (41) reported that silencing of ACA1-077100, which encodes XP_004335869.1 annotated as Rab1/RabD family small GTPase, disabled phagosome formation in *A. castellanii* fed with *Escherichia coli*. Currently, there are no reports on the downstream effectors recruited by *Acanthamoeba* Rabs for phagosome maturation and delivery to lysosomes. Our ongoing research aims to address these questions, and the use of plasmids and biological assays from the present study, along with recently developed CRISPR-Cas9 technology-based genetic manipulation protocols (108), should be valuable tools for fostering research on the cellular microbiology of *Acanthamoeba*-bacteria interactions.

## MATERIALS AND METHODS

### *Acanthamoeba castellanii* Neff cultivation and transfection

The reference *A. castellanii* strain Neff (7, 45) (Table 1) was purchased from the American Type Culture Collection (ATCC 30010) and cultured at 30°C in 25-cm$^2$ cell culture flasks (Nunc) with 7-mL peptone–yeast extract–glucose (PYG) medium (https://www.atcc.org/products/30010). Stock cultures were grown to approximately 80%–90% confluency, as determined by phase-contrast microscopy using an inverted microscope (AxioVertA1, Zeiss) equipped with an LD Plan-A 10× objective and the Fiji (109) PHANTAST plugin (110), harvested by centrifugation at 500 × *g*, washed once in 1 vol PYG, resuspended in 3-mL PYG, and stored at 4°C until required (111). Stable transfection of *A. castellanii* Neff with selected plasmids was performed in six-well plates (Nunc) using SuperFect Transfection Reagent (Qiagen) and column-purified plasmids (ZymoPURE II Plasmid Midiprep kit, Zymo Research) following the protocol reported by Peng et al. (69).

### Bioinformatic methods to identify Rab7 GTPase homologs in the *A. castellanii* strain Neff RefSeq (ACA1) and Neff-2 proteomes

We constructed a set of 94 manually curated small Rab GTPase sequences from *Homo sapiens* (*n* = 62; Rab1 to Rab40, excluding the large Rab-like GTPases Rab44, Rab45, and Rasef), *Saccharomyces cerevisiae* (*n* = 10), and *Dictyostelium discoideum* (*n* = 22), retrieved from UniprotKB/Swis-Prot v2022_04 (112). We refer to this set as Rab_SPhyd94 and use it as the gold standard for canonical small Rab GTPases from the phyla Opisthokonta (animals and fungi) and Amoebozoa (113). We used phmmer from the HMMER 3.3.2 suite (114) and the *H. sapiens* RAB7A_HUMAN (P51149) and *D. discoideum* RAB7A_DICDI (P36411) proteins as the query sequences to search for Rab7 homologs in the *A. castellanii* strain Neff reference (RefSeq) proteome (GCF_000313135.1; v2021-10-09) as well as in the recently reported chromosome-level assembly (Neff-2) and annotation for this strain (79), imposing an E-value cutoff threshold of 1e-35. The Neff-2 assembly was retrieved from https://zenodo.org/record/6800059 as it has not been deposited in public sequence databases. The *Acanthamoeba* phmmer hits were scanned with FIMO (71) from the MEME suite v5.4.1 (115) to identify sequences containing at least two of the five Rab family-specific motifs (RabF1-RabF5), imposing a *P*-value cut-off of 5e-4 (72). RabF hidden Markov models were retrieved from the rabifier2 (73) GitHub repository https://github.com/evocell/rabifier. The resulting data set was further filtered by selecting

sequences containing one or two cysteine residues among the five C-terminal amino acids that serve as prenylation sites (74, 75). Finally, orthologs between the Rab_SPhyd94 reference set and the two *A. castellanii* Neff proteomes were computed using easy-rbh from the MMseqs2 package (116).

## Domain composition scanning, structure prediction, pairwise structure alignment, structure-guided multiple sequence alignment, and phylogenetic analysis of Rab GTPase homologs

Domain composition scanning of all Rab homologs was performed using InterPro-Scan v5.61-93 (117) against the InterPro database (118). We used ColabFold's "Alpha-Fold2_advanced" notebook (94) with max_recycle = 6 and MMseqs2 (116) alignments to generate AlphaFold2 (93) *de novo* tertiary structure predictions for Rab protein sequences not available in the AlphaFold Protein Structure Database (119) and ranked them according to their predicted local distance difference test value. Alignments between pairs of Rab 3D structures in PDB format were generated using TM-align (120), evaluating their global superposition with the TM-score metric (120). Individual 3D protein structures and protein structure alignments were rendered using PyMOL v2.5.0 (121) for Linux. Multiple sequence alignments of Rab homologs were generated with MAFFT v7.505 compiled on a GNU/Linux machine, enabling the dash_client option (version 1.1) and invoking mafft with the "--dash" flag for structure-guided multiple sequence alignment (122) together with the "--localpair --maxiterate 1000" options for the highest accuracy (123). We used ESPrit3 (78) to map secondary structural elements from the human Rab7A structure (PDB code 1t91) to the mafft-dash alignment. Curated multiple sequence alignments were subjected to maximum-likelihood phylogenetic analysis under the best-fitting substitution model selected with ModelFinder (124) using the Bayesian information criterion, as implemented in IQTree2 v2.2.0 (125). Bipartition bootstrap support values were computed using UFBoot2 (126).

## Construction of *Acanthamoeba* expression plasmids to generate C- and N-terminal fusions to fluorescent proteins

We used the empty backbone mammalian expression plasmids mEGFP-C1, mEGFP-N1, mCherry2-C1, and mCherry2-N1, a gift from Michael Davidson (AddGene plasmids 54759, 54767 54563, and 54517; Table 1), to construct the empty backbone expression plasmids pAcaGAPDH_mEGFP-C1, pAcaGAPDH_mEGFP-N1, pAcaGAPDH_mCherry2-C1, and pAcaGAPDH_mCherry2-N1 (Table 1). The cloning strategy involved two steps: (i) The SV40 promoter was removed from the four mammalian expression plasmids by Q5 high-fidelity polymerase (NEB) PCR amplification with primers neo-Kozak_SpeI.F/pBABE_PacI.R (4,407-bp amplicon), which reads outward from the SV40 promoter region (Table 2), and replaced with the *A. castellanii* Neff TBP promoter region (198 bp) (53) and amplified from genomic DNA using Phusion polymerase (Thermo Fisher Scientific) and primers TBP-110_PacI.F/TBP + 68_SpeI.R (Table 2). The TBP and vector amplicons were digested with *Spe*I + *Pac*I (NEB) and ligated with T4 DNA polymerase (Thermo Fisher Scientific) to generate pAcaPrTPB_bkbn_mEGFP-C1, pAcaPrTPB_bkbn_mEGFP-N1, pAcaPrTPB_bkbn_mCherry2-C1, and pAcaPrTPB_bkbn_mCherry2-N1 (Table 1). (ii) The CMV enhancer and promoter regions were removed from the latter plasmids by *Ase*I + *Nhe*I (NEB) double digestion and replaced with the *A. castellanii* Neff GAPDH promoter region, as defined by Bateman (48). The GAPDH promoter region was amplified from genomic DNA with Phusion-polymerase using primers Aca_Pr_GAPDH.F2 and Aca_Pr_GAPDH.R (Table 2). The resulting amplicons were digested with *Ase*I + *Nhe*I and ligated with double-digested plasmid DNAs to generate the *Acanthamoeba* backbone expression vectors pAcaGAPDH_mEGFP-C1, pAcaGAPDH_mEGFP-N1, pAcaGAPDH_mCherry2-C1, and pAcaGAPDH_mCherry2-C1 (Table 1). *Escherichia coli* DH5α was used as the host for all cloning experiments. Transformants harboring recombinant plasmids were grown in LB medium at 37°C with appropriate antibiotics.

## Amplification, sequencing, and cloning of ACA1_074580 encoding Rab7A protein

The *A. castellanii rab*7A gene ACA1_074580 was PCR amplified using Phusion DNA polymerase (Thermo Fisher Scientific) from the genomic DNA purified from *A. castellanii* Neff (ATCC 30010) trophozoites using the DNeasy UltraClean Microbial Kit (Qiagen) and primers ACA_rab7a.F/ACA_rab7a.R2 (Table 2). The amplicon was sequenced twice from both ends by Sanger sequencing at our university sequencing facility (IBt-UNAM, Mexico), and the reads were assembled into a contig using Phrap (127). Intron splicing sites were determined by aligning the RAB7A_HUMAN protein (P51149) to the full-length gene sequence using minipot v0.7 (80). The amplicon was digested with *Bgl*II and *Bam*HI and cloned into pAcaGAPDH_mEGFP-C1, yielding pAcaGAPDH_mEGFP-C1_rab7A of 5,531 bp (Table 1).

## Generation and FACS enrichment of an *A. castellanii* line stably overexpressing an mEGFP-Rab7A fusion protein

The pAcaGAPDH_mEGFP-C1_rab7A plasmid was used to transfect *A. castellanii* trophozoites using SuperFect (Qiagen). Adherent cells were transfected as described by Peng et al. (69) with minor modifications. Briefly, 4 µg (~0.6 µg/mL) of the column-purified plasmid (EndoFree Plasmid Maxi Kit, Qiagen) was diluted in 100 µL of PYG, gently mixed with 20-µL SuperFect, and incubated for 5 min at room temperature. The DNA-Superfect complex was then added to exponentially growing Neff cells at a density of ~75% confluence in six-well tissue culture-treated polystyrene plates. After a 3-h incubation at 30°C, the cells were washed with phosphate-buffered saline (PBS) and incubated for an additional 3 h with the mixture. The cells were then washed with pre-warmed PYG and allowed to recover in fresh PYG for 3 h before adding G418 at 12.5 µg/mL. After 24 h, the medium was replaced with PYG-G418 (25 µg/mL) every third day until adherent and active trophozoites were detected by standard bright field microscopy after 5–7 days. Non-transfected Neff trophozotites were used as controls. The proportion of mEGFP-RAB7A positive cells was determined using a BD FACSCanto II Flow Cytometer at the Cytometry Unit from IBt-UNAM (UcySC-PCTCM), and the population of fluorescent cells enriched by two consecutive passes of trophozoite suspensions through this fluorescence-activated cell sorter in PBS (pH 7.4) + 0.53 mM EDTA at ~1 × $10^6$ cfus/mL. The enriched fluorescent population was cultivated statically in PYG-G418 (50 µg/mL) in 25-mL vented culture flasks at 30°C until it reached ~80% confluence. The cells were resuspended in PYG medium containing 10% (vol/vol) ice-cold dimethyl sulfoxide (DMSO), aliquotted into 1.8-mL cryo-tubes on ice, and frozen in a cryo-box at −80°C.

## Construction of new mTn7 delivery vectors for stable tagging of *Stenotrophomonas maltophilia* with a constitutively expressed mScarlet-I cassette

Plasmids pUC18T-mini-Tn7T and pFCM1 (70) were used as the building blocks to construct pUC18T-mTn7TC1. Briefly, the pUC18T-mini-Tn7T backbone was PCR amplified using Q5 polymerase (NEB) with primers mTn7_SPPfw/mTn7_*Sph*I.Rev (Table 2), followed by digestion of the 3,554-bp amplicon with *Swa*I and circularization by self-ligation with T4 DNA ligase to yield pUC18T_mTn7T_SSPP (Table 1). The Cm$^R$ cassette from pFCM1 (70), along with the flanking FRT sites, were PCR amplified with primers M13pUC_*Swa*I.R and M13pUC_*Pac*I.F (Table 2), cloning the resulting 1,329-bp amplicon as a *Pac*I-*Swa*I fragment into the corresponding sites of pUC18T_mTn7T_SSPP, generating pUC18T-mTn7TC1 (Table 1). This plasmid was digested with *Spe*I and *Eco*RI to clone an 886-bp fragment from pMRE-Tn7-145 (128), amplified with primers pMRE-Tn7-145_SDmScarlet.F and pMRE-Tn7-145_SDmScarlet.R (Table 2), which contained on one side the mScarlet-I gene with a strong Shine-Dalgarno (TGGAGGA) sequence, located 7 bp upstream of the start codon, and a lambda terminator on the other side, yielding pUC18T_mTn7TC1_Pr_mScarlet-I (Table 1). Finally, a 271-bp

fragment containing the strong constitutive Pc promoter from class 1 integrons (68) was PCR amplified from pMRE-Tn7-145 (128) using primers pMRE-Tn7-145_Pc.F and pMRE-Tn7-145_Pc.R, which was directionally cloned upstream of the mScarlet-I gene (67) in pUC18T_mTn7TC1_Pr_mScarletI as a *Pac*I-*Spe*I fragment, yielding pUC18T_mTn7TC1_Pc_mScarlet-I of 5,893 bp (Table 1).

## Tagging of *S. maltophilia* Sm18 with constitutively expressed mScarlet-I using the new mTn7 delivery vector pUC18T_mTn7TC1_Pc_mScarlet-I

The environmental *S. maltophilia* strain Sm18 (54) was made electrocompetent using the rapid microcentrifuge-based protocol reported by Choi et al. (129). Fifty nanograms of the delivery plasmid were mixed with 50 ng of pTNS2 (70) and 100 µL of electrocompetent cells at room temperature, and the mixture was transferred to a 2-mm gap-width electroporation cuvette (BioRad). After applying a pulse (25 µF; 200 Ω; 2.5 kV) on a Bio-Rad Gene Pulser Xcell (Bio-Rad), 1 mL of room temperature LB medium was added to each cuvette, and the cells were transferred to a glass tube for recovery for 2 h at 30°C in a rotary shaker without antibiotics. The cells were then harvested by centrifugation, the supernatant discarded, and the cell pellet resuspended in the residual medium before plating on LB + gentamycin (Gm20) and chloramphenicol (Cm30) plates (antibiotic quantities in micrograms per milliliter). The cells were incubated at 30°C until colonies appeared (24–48 h). Fluorescent colonies growing on LB plates were identified using a Typhoon FLA 9500 laser scanner (GE Healthcare Life Sciences) with appropriate laser and filter settings. To verify that the insertions were located downstream of the *glmS* gene, we designed primer Smal_glmS_down_1549F (Table 2), which binds to a conserved region of the gene in *S. maltophilia*, and used it together with primer PTn7R (70) (Table 2). Sequencing of the resulting amplicon revealed that mTn7 was inserted 25 nucleotides downstream of the single-copy *glmS* gene in the expected orientation, with the Tn7R border facing the *glmS* 3′-end [(glmS_TAAstop)CACCCGTCAGCTCGGTTGGAG GTTG(Tn7R_TGTGGG...)]. We selected the positive strain Sm18::mTn7TC1_Pc_mScarlet-I (Table 1) for the downstream experiments.

### Evaluation of mTn7 insertion stability and mutant fitness

The stability of the mTn7 insertion in strain Sm18::mTn7TC1_Pc_mScarlet-I was evaluated by continuous subculturing for 14 days in 5 mL of LB without antibiotics, checking for loss of Cm$^R$ by parallel streaking of 50 colonies on LB and LB-Cm30 plates, and fluorescence was evaluated using a Typhoon FLA 9500 laser scanner (GE Healthcare Life Sciences) after 7 and 14 days of subculturing, revealing that 100% of the colonies (*n* = 100) were Cm$^R$ and fluorescent after days 7 and 14 of the experiment (data not shown). To assess the relative fitness of the parental and tagged strains, we compared their growth curves in 96-well non-treated polystyrene plates (Costar 3370) filled with 200-µL LB and incubated at 30°C and 37°C with orbital shaking at 220 rpm, using an Epoch two microplate spectrophotometer (BioTek). Six independent overnight pre-cultures of each strain were prepared in LB at 30°C and 37°C, diluted to an OD$_{600}$ ≈0.20 in fresh LB pre-warmed at the indicated temperatures, incubated for an additional 2 h at the corresponding temperatures, and diluted again to an OD$_{600}$ ≈0.10 before inoculating 180 µL of LB with 20 µL of the bacterial suspension in each well. We used eight technical replicates for each of the five biological replicates and measured the OD$_{600}$ every 15 min. Growth curve parameters were estimated using the R package GrowthCurver (130), and the means of growth parameters between the parental and mutant strains were compared using the non-parametric Wilcoxon test, as implemented in ggpubr (131).

## Quantification of biofilm formation by *S. maltophilia* Sm18 and Sm18::mTn7TC1_Pc_mScarlet-I

Overnight cultures were incubated in LB at 30°C and 37°C in a rotary shaker at 200 rpm, followed the following morning by dilution of the inocula in fresh, pre-warmed LB to

an $OD_{600}$ ≈0.10 before inoculating 180 µL of LB in each well of 96-well non-treated polystyrene plates (Costar 3370) with 20 µL of the bacterial suspension. Eight technical replicates were used for each of the five biological replicates of each strain. The plates were statically incubated for 48 h at 30°C and 37°C in a humid chamber. Immediately before biofilm biomass quantification, the optical density of each well was evaluated ($OD_{620}$) using an Epoch two microplate spectrophotometer (BioTek), followed by CV (0.1% in water) staining of the biofilm for 15 min using standard procedures (132). After washing off (3×) excess dye and fixing at 60°C for 1 h, CV was extracted with 200 µL of 30% (vol/vol) acetic acid in water well$^{-1}$ and quantified at 550 nm using an Epoch two instrument. Biofilm formation ($OD_{550}$) was normalized to cell growth ($OD_{620}$) in each well, and the means between treatments were compared using the non-parametric Wilcoxon test, as implemented in ggpubr (131).

### *Acanthamoeba castellanii*–*Stenotrophomonas maltophilia* Sm18 co-culture for live-cell imaging

Co-cultures were set up by first seeding 20-mm diameter glass bottom (0.17-mm thick coverslip) cell culture (treated) dishes (NEST 801001) with $5 \times 10^5$ CFUs of actively growing *A. castellanii* strain Neff trophozoites, previously quantified with the aid of a Neubauer counting chamber (Blaubrand, Merck), which were incubated at 30°C for 1 h in a 1.6-mL salt solution of the defined MM-MOPS medium (133) plus 0.8% (wt/vol) D-glucose (MMsalts-MOPS-Glc, pH 7.2). *Stenotrophomonas maltophilia* Sm18 (54) was grown in LB medium at 30°C. Overnight cultures were washed once in fresh LB, diluted to an $OD_{600}$ ≈0.7, and incubated at 30°C until reaching an $OD_{600}$ ≈1.0, equivalent to ~$4.8 \times 10^8$ CFUs/mL. Dilutions of the Sm18 culture were quantified using a Neubauer counting chamber, and 100-µL suspensions at the proper concentration of cells were added to each cell culture dish to reach the desired multiplicity of infection (MOI 12.5, MOI 25, MOI 50, and MOI 100) and centrifuged at $500 \times g$ for 5 min at 30°C to synchronize phagocytosis.

### LCI using widefield fluorescence microscopy

LCI of *A. castellanii* Neff lines in co-culture with *S. maltophilia* Sm18 cells stained with various fluorescent dyes or expressing mScarlet-I was performed on an AxioVertA1 inverted microscope (Karl Zeiss) equipped with an HXP Compact Light Source containing a mercury short-arc reflector lamp, a long-distance LD Plan-Neofluar 63×/0.75 Corr Ph2 M27 objective (LDPNF63×), an EC Plan-NeoFluar 63×/1.25 oil immersion objective (ECPNF63×), filter sets FS38 (BP 470/40, FT 495, BP 525/50; for mEGFP), FS43 (BP 545/25, FT 570, BP 605/70; for pHrodo Red, Rhodamine and mScarlet-I), and FS50 (BP 640/30, FT 660, BP 690/50; for pHrodo-Deep red), an Axiocam 202 monochromatic camera, and Zen Lite software, all from Karl Zeiss. Metadata for all images were saved in czi format and processed with Fiji (109), making use of the Bio-Formats (134) and Quick-Figures (135) plugins for figure editing. Cells were grown in 20-mm diameter cell culture-treated dishes with 0.17-mm thick cover-glass bottom (NEST 801001) and imaged with the ECPNF63× objective or in six-well polystyrene tissue culture-treated plates (NEST 703001) and imaged with the LDPNF63× objective. Trophozoites were immobilized immediately before imaging by adding a 2- to 3-mm-thick block of 0.25% GelRite (Sigma-Aldrich) in MMsalts-MOPS-Glc with 0.05% $CaCl_2$.

### LCI of *Acanthamoeba* macropinocytic and phagocytic pathways with fluorescent dextran and pH-sensitive BioParticles

We used a neutral dextran-rhodamine B conjugate (10,000 MW) from Invitrogen as a fluorescent fluid-phase tracer for macropinocytosis (136). pHrodo Deep Red Zymosan BioParticles Conjugate (Invitrogen) was used as a no-wash, pH-sensitive phagocytosis tracer. These are non-fluorescent outside the cell at neutral pH but fluoresce brightly in acidic environments, such as endocytic vesicles and lysosomes. We added 200 µL

of a 10× solution of dextran-rhodamine B or Zymosan pHrodo BioParticles solution in MMsalts-MOPS-Glc (pH 7.2) to 20-mm cell culture dishes with glass bottom (0.17-mm-thick coverslip) cell culture (treated) dishes (NEST 801001) previously seeded with $5 \times 10^5$ trophozoites, followed by centrifugation at $800 \times g$ for 5 min at 30°C to synchronize endocytosis. LCI was performed at different time points, between 30 and 120 min. Dishes treated with dextran-rhodamine B conjugate were washed twice with fresh MMsalts-MOPS-Glc (pH 7.2) at 30°C immediately before imaging to remove extracellular particles and reduce the background signal.

### *Acanthamoeba* phagosome acidification and uptake kinetics of live *S. maltophilia* Sm18 cells labeled with pHrodo red succinimidyl ester

We used a Synergy H1 microtiter plate reader (BioTek) to follow the fluorescence readout over time emitted by live *S. maltophilia* Sm18 cells stained with amine-reactive pHrodo Red succinimidyl ester dye (Invitrogen) in 96-well, flat-bottom, clear tissue culture-treated surface black polystyrene microplates (Corning CLS3603). Briefly, a 10 mM stock solution of pHrodo Red dye was prepared in anhydrous DMSO. Sm18 cells ($3 \times 10^9$) grown in LB at 30°C were harvested at the late exponential phase ($OD_{600} = 1$), washed twice in PBS (pH 7.4), and resuspended in 1 mL of 100 mM $NaHCO_3$ (pH 8.5). We used 190 µL of the bacterial suspension (5.4e + 08 CFU/mL) for staining with 5 µL of 10 mM pHrodo stock solution. The mixture was incubated for 30 min at room temperature in the dark, with gentle agitation. The stained cells were washed (three times) with 1-mL PBS to remove the free dye and resuspended in 200 µL of MMsalts-MOPS-Glc. Each well was seeded with 1e + 5 trophozoites in 100 µL of MMsalts-MOPS-Glc and incubated for 1 h at 30°C to allow trophozoites to adhere. Stained Sm18 cells were counted in a Neubauer chamber (Blaubrand, Merck), and the suspension was adjusted to reach the target MOI. Phagocytosis was synchronized by centrifugation at $800 \times g$ for 3 min at 30°C. The microtiter plate was incubated statically at 30°C in a plate reader at 559 nm/585 nm (excitation/emission) wavelengths, setting the optics to read from the bottom of the plate at low lamp energy with 20 measurements per data point. Wells inoculated with live-stained Sm18 cells in MMsalts-MOPS-Glc served as controls for background subtraction.

### Quantitative evaluation of the intracellular replication of *S. maltophilia* using a microtiter plate assay

To determine the intracellular replication capacity of Sm18::mTn7TC1_Pc_mScarlet-I in *Acanthamoeba* trophozoites at different MOIs, we implemented a microtiter plate assay that measured replication as the increase in mScartlet-I fluorescence emitted by co-cultures of amoebae and the tagged bacterial strain. Fluorescence readout was recorded using a Synergy H1 plate reader (BioTek). Each experimental well was seeded with $1 \times 10^5$ trophozoites in 100 µL of MMsalts-MOPS-Glc and incubated at 25°C for 30 min to allow for cell attachment. The medium was replaced with Sm18::mTn7TC1_Pc_mScarlet-I cells in 100 µL of MMsalts-MOPS-Glc, adjusted to the target MOI. Phagocytosis was synchronized by centrifugation at $800 \times g$ for 3 min at 25°C. The plates were immediately placed in a plate reader and incubated statically at 25°C, and red fluorescence was measured every 30 min for 48 h at 568 nm/594 nm (excitation/emission) wavelengths. The optics of the instrument were set to read from the bottom of the plate at low lamp energy, with 20 measurements per data point. Wells containing only MMsalts-MOPS-Glc medium, seeded with or without trophozoites or Sm18::mTn7TC1_Pc_mScarlet-I were used as controls. In addition, wells containing filtered (0.24 µM, Millipore) "conditioned" medium from 24-h-old Neff cultures were used to evaluate the growth of Sm18::mTn7TC1_Pc_mScarlet-I on the Neff secretions.

## ACKNOWLEDGMENTS

We gratefully acknowledge the technical assistance and support provided by Dr. Yvonne Rosenstein and MsC Erika Melchy from the UcySC-PCTCM cytometry unit and colleagues at the DNA synthesis and sequencing unit of IBt-UNAM. We thank Alfredo Hernández and Víctor del Moral for their expert technical support with server maintenance at Centro de Ciencias Genómicas, UNAM. Dr. Luis Cárdenas from IBt-UNAM is thanked for providing materials and Dr. Ayari Fuentes from CCG-UNAM for providing access to the multimodal plate reader.

We gratefully acknowledge the financial support received from Consejo Nacional de Humanidades Ciencias y Tecnologías (CONAHCyT Mexico, A1-S-11242) and Programa de Apoyo a Proyectos de Investigación e Innovación Tecnológica (PAPIIT), Universidad Nacional Autónoma de México (DGAPA-PAPIIT, UNAM: IN209321). J.C.V.N., D.M.V.E., and F.S.A.Z. were recipients of CONAHCyT scholarships 745705, 720071, and 659308, respectively.

## AUTHOR AFFILIATIONS

[1]Centro de Ciencias Genómicas, Universidad Nacional Autónoma de México (UNAM), Cuernavaca, Morelos, Mexico
[2]Programa de Doctorado en Ciencias Biomédicas, UNAM, Mexico City, Mexico
[3]Programa de Maestría y Doctorado en Ciencias Bioquímicas, UNAM, Mexico City, Mexico

## AUTHOR ORCIDs

Julio C. Valerdi-Negreros  http://orcid.org/0000-0001-7536-5483
Pablo Vinuesa  http://orcid.org/0000-0001-6119-2956

## FUNDING

| Funder | Grant(s) | Author(s) |
| --- | --- | --- |
| Consejo Nacional de Humanidades Ciencias y Tecnologías (CONAHCYT) | A1-S-11242 | Pablo Vinuesa |
| UNAM \| Dirección General de Asuntos del Personal Académico, Universidad Nacional Autónoma de México (DGAPA) | IN209321 | Pablo Vinuesa |

## AUTHOR CONTRIBUTIONS

Javier Rivera, Data curation, Investigation, Methodology, Validation | Julio C. Valerdi-Negreros, Data curation, Formal analysis, Investigation, Methodology, Validation | Diana M. Vázquez-Enciso, Investigation, Methodology | Fulvia-Stefany Argueta-Zepeda, Investigation, Methodology | Pablo Vinuesa, Conceptualization, Data curation, Formal analysis, Funding acquisition, Investigation, Methodology, Project administration, Resources, Supervision, Validation, Visualization, Writing – original draft, Writing – review and editing

## ADDITIONAL FILES

The following material is available online.

Open Peer Review

**PEER REVIEW HISTORY (review-history.pdf).** An accounting of the reviewer comments and feedback.

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
