## [Reviewer comments · Microbiology Spectrum]

Microbiology Spectrum

Phylogenomic, structural, and cell-biological analyses reveal that *Stenotrophomonas maltophilia* replicates in acidified Rab7A-positive vacuoles of *Acanthamoeba castellanii*

Javier Rivera, Julio Valerdi-Negreros, Diana Vázquez-Enciso, Fulvia-Stefany Argueta-Zepeda, and Pablo Vinuesa

Corresponding Author(s): Pablo Vinuesa, Universidad Nacional Autonoma de Mexico - Campus Morelos

Review Timeline:

Submission Date:	August 1, 2023
Editorial Decision:	September 29, 2023
Revision Received:	January 8, 2024
Accepted:	January 15, 2024

Editor: Michael Ginger

Reviewer(s): Disclosure of reviewer identity is with reference to reviewer comments included in decision letter(s). The following individuals involved in review of your submission have agreed to reveal their identity: Monica Crary (Reviewer #1); Sutherland K Maciver (Reviewer #3)

Transaction Report:

DOI: <https://doi.org/10.1128/spectrum.02988-23>

September 29, 2023

Prof. Pablo Vinuesa
Universidad Nacional Autonoma de Mexico - Campus Morelos
Centro de Ciencias Genómicas
Av. Universidad s/n
Col. Chamilpa
Cuernavaca, Mor. 62210
Mexico

Re: Spectrum02988-23 (**A new plasmid toolkit reveals that *Stenotrophomonas maltophilia* replicates in acidified Rab7A-positive vacuoles of *Acanthamoeba terricola***)

Dear Prof. Pablo Vinuesa:

Thank you for submitting your manuscript to Microbiology Spectrum. Your work has received positive comments from three reviewers with relevant expertise. However, there are a number of critical comments that mean your manuscript needs some revision before it should be acceptable for publication. The reviewers' comments are provided at the end of this decision letter and in the two attachments. In addition, one of the reviewers in their comments to me has questioned whether the title of your manuscript is fully reflective of your study. I agree with the referee's view and encourage you to consider a new title that fully aligns with the work described. Finally, there is also a sense that it would be possible to condense the length of your manuscript at least a little.

Link Not Available

Sincerely,

Michael Ginger

Journals Department
Reviewer comments:

Reviewer #1 (Comments for the Author):

Figure 3. Not sure why they haven't pseudo-colored the images. Images seem out of focus. For amoeba, a z-projection may have been better. No comments about the increased in vesicles in Rab7 overexpression?

Fig 4 Missing arrow heads?
Why not merge brightfield of EFH?
Are these scales right? Is scale different for H? It doesn't look like it matches?

Figure 5. I get how annoying burning a scale in but I think you need to pick one and stick to it.

I'm struggling with it being a Toolbox when the transfection rate is 5-10%. You are going to struggle to study anything related to the pathway or dynamics beyond a couple of images with that transfection rate. It is not a major improvement over what is currently in the literature. You've done a lot of plasmid work for both organisms here. Why that *S. maltophilia* specifically? would your plasmid work with others?

Also, using Neff (beyond the genomes that are available) is not super representative of the genus as a whole. Did you do anything to the Neff culture-wise to make it more "normal" versus an organism that has been axenically cultured for 60 years?

I don't know how much you extended the understanding of the Rab/Ras pathway in this paper beyond suggesting it co-localizes with intracellular bacteria. Has it really been established that *S. maltophilia* is only in acidified phagosomes?

So it is pretty clear visually that *Acanthamoeba* is not super thrilled with mEGFP-Rab7A given the overwhelming amount of vesicles in the amoeba compared to mEGFP alone. I didn't see any discussion of that mentioned or the downstream effects of your experiment where the amoeba is clearly over producing vesicles and then you are making conclusions about the type of vesicles you are finding the bacteria in.

There's a bit of a cohesiveness issue with the paper. You made plasmids that labeled bacteria (again, not sure why a commercial plasmid wouldn't have worked; either way, most of that activity could be moved to the methods). *Acanthamoeba* you made plasmids, the transfection rate is very low (as is found with the other plasmids for amoeba so I'd be careful about suggesting some improvement). Your plasmid clearly had a phenotypic impact that isn't addressed and needs to be. You found a bunch of Rab genes in Neff, picked one, and made a plasmid against it. You don't actually prove its an overexpression or that the protein is functional within amoeba, just that it is tagged and seems localized to the vesicle. This seems like a few papers together as one. I think you could focus on what you learned about bacteria infection in amoeba and maybe move some things into the methods rather than making it a bulk of the paper.

Reviewer #3 (Comments for the Author):

This study investigates the intracellular location of replication of an intracellular pathogen *Stenotrophomonas maltophilia* in *Acanthamoeba*, a facultative human pathogen. GFP- Rab7 GTPase fusion proteins are used to delineate organelle distribution. The work is of high quality and represents a significant step forward in the field and offers new tools to unlock the cell biology of *Acanthamoeba*. The paper is very well written and very clear. The main importance of this paper is the tools that it offers to other researchers, but the contents are extremely interesting also! There are a few relatively minor points which need to be addressed however.

Line 30 "amoebae" plural not "amoeba" singular.

Line 33. Perhaps "uncharacterized" would be better than "marginal" here as it is our current knowledge of the Rab GTPases that is marginal not them themselves?

Line 35. Yes, the current classification of genus *Acanthamoeba* is not complete or satisfactory, and it is almost certain that in due course a better nomenclature will be proposed after consultation across the interested parties. Whereas it seems unlikely that the species name *castellanii* would be retained, it is unhelpful at this stage to rename the Neff strain. Species name *terricola* has taxonomic problems also, so its probably better to stick to the *Acanthamoeba castellanii* (Neff) description that the field understands until a complete and agreed system is established. Work on this problem is ongoing. Calling the Neff strain *terricola* and then having to change this again in a short time would cause unnecessary confusion.

Note that throughout the current manuscript the name *castellanii* is still being used in a few places (e.g. Lines 270, 722, 901, & 910).

More details about Scarlet are needed. It should be first of all stated that it is an RFP derivative. Other required details include is it pH sensitive?

Line 76. A small point but the references 7 and 8 are not a good match to support the argument that *Acanthamoeba* was adopted early as a model eukaryote. These references are dated 2013 and 2022? Instead I suggest the following early paper in which the now ubiquitous Myosin 1 motor protein was isolated from *Acanthamoeba* as an example of the utility of the amoeba as a eukaryotic model?

Pollard TD, Korn ED (1973) *Acanthamoeba* myosin I. isolation from *Acanthamoeba castellanii* of an enzyme similar to muscle

myosin. J Biol Chem 248:4682-4690.

Line 116. "73% of the eleven Rab7 homologs"? Do the authors mean "8 of the 11 Rab7 homologs"?

Line 414. The "non-characterized, non-lytic, exocytic pathway" show similarities to the "vomocytosis" phenomenon (Seoane, P. I., & May, R. C. (2020). Vomocytosis: what we know so far. Cellular Microbiology, 22(2), e13145.). Have the authors considered this possibility?

Figure 4, 5 and 6. the panels showing the amoebae are labelled "bright field" but the images are clearly phase contrast?

Figure 5. It seems odd that the vacuoles containing *Stenotrophomonas maltophilia* labelled with pHrodo are uniformly red with no indication of the expected rod shape that the bacterium has? Is it possible that the label has become detached from the bacteria? Have the bacteria been digested? Is there another explanation? This apparent problem is also seen with the RFP scarlet expressing bacteria in Figure 7.

The expected rod shapes can be seen in Figure 7B but not actually in the cells or vesicles. This Part B of Figure 7 is very small, it contains important information and should be shown full size? The fluorescence is used as a proxy for bacterial numbers. These experiments show that the bacteria continue to produce the RFP but can the authors make a strong claim that this reflects the actual intracellular growth of the bacteria? Rod-shaped Bacteria are clearly visible outside the amoebae but the vacuoles appear to be empty by phase contrast? (E.M. would sort this out but this may not be feasible for the authors to perform). This is important as this is the main message from the paper.

Staff Comments:

Preparing Revision Guidelines

Please return the manuscript within 60 days; if you cannot complete the modification within this time period, please contact me. If you do not wish to modify the manuscript and prefer to submit it to another journal, please notify me of your decision immediately so that the manuscript may be formally withdrawn from consideration by Microbiology Spectrum.

A. castellanii, reclassified as *A. terricola*, is a free-living amoeba with clinical and environmental significance. Despite being a valuable model organism, it is not as well-studied or utilised as other amoebae, such as *Dictyostelium discoideum* due to the limited availability of molecular biology tools. In this manuscript, Rivera et al. introduce a new set of plasmids that can be used to transfect *A. castellanii*, enabling the creation of fusions both upstream and downstream of fluorescent proteins. Using these plasmids, they were able to study the localisation of the GTPase ortholog Rab7A in *A. castellanii* and determine that it localises to acidified vacuoles. The authors also conducted infection assays with *S. maltophilia* and found that it is resistant to *A. castellanii* predation by forming a SmCV inside the amoeba, allowing it to replicate in this niche.

General comments:

- I thoroughly enjoyed reading this manuscript and think it is very well-written. The Abstract and Importance sections were clear and concise, providing valuable information without being repetitive. The Introduction was succinct yet informative, including necessary background information and a thorough literature review.
 - While I appreciate the paragraph discussing the reclassification of *A. castellanii* to *A. terricola*, I do not believe it is relevant to this particular work. I am concerned that the new name in the title may cause confusion for some readers and potentially cause the article to miss its intended audience. I suggest acknowledging the reclassification and clarifying that *A. castellanii* will be used throughout the article for simplicity.
 - It is evident that a great deal of work went into this manuscript and while I know that projects do not always/almost never proceed linearly, it appears to me that two separate stories are being merged together. Nonetheless, I still believe the manuscript is well-written, and I am unsure what changes could be made to improve it.
1. This is probably the obvious question, but why didn't you use the pGAPDH-EGFP as the backbone? As stated in the manuscript, it is the most common plasmid used to transfect *A. castellanii*, it's simpler to change one fluorescence gene than two promoters, and pGAPDH-EGFP also has an Amp^R resistance gene for selection in *E. coli* when cloning. Regarding the cloning limitations of pGAPDH-EGFP, although the sequence is not available, it can be sequenced, and there are other tools to cut or insert fragments that don't require restriction sites/enzymes. Regarding promoter choice, Bisio et al. 2023 (<https://doi.org/10.1038/s41467-023-36145-4>) is an appropriate reference.
Thank you for making the plasmids available on Addgene, it is an important practice.
 2. Regarding the transfection of *A. castellanii*, did you use G418 to select for positive transformants and keep the line, or only FACS after transfection? How long did you keep the line for after transfection? If you passaged the culture after FACS, could you comment on plasmid/transfection stability without using selection?
 3. The phylogenetic analysis was thorough and well-explained, and the results were exciting. Kudos!
 4. I found LCI and the two assays to characterise the localisation of Rab7A to be very fitting and creative. The micrographs were illustrative.
 5. Transforming environmental strains is not a trivial task, so this is notable work, and I appreciate the creative use and adaptation of established protocols.
 6. I'm curious, why did you perform the infections at 25°C instead of 30°C as in previous assays?
 7. The estimation of intracellular replication of *S. maltophilia* was an excellent addition.

Notes:

- Write the long form of "ppm" at least once.
- Figure 3 (C) legend: "firebrick" might not be understood by everyone as being a colour.
- Typos: Line 190 (Ran GTPase), Line 309 (*A. terricola*/*S. maltophilia*) vs Line 495 (*Acanthamoeba-Stenotrophomonas*), Line 540 (Gm^R), Line 853 (wright) Line 925 (AlphaFold2).

This study investigates the intracellular location of replication of an intracellular pathogen *Stenotrophomonas maltophilia* in *Acanthamoeba*, a facultative human pathogen. GFP- Rab7 GTPase fusion proteins are used to delineate organelle distribution. The work is of high quality and represents a significant step forward in the field and offers new tools to unlock the cell biology of *Acanthamoeba*. The paper is very well written and very clear. The main importance of this paper is the tools that it offers to other researchers, but the contents are extremely interesting also! There are a few relatively minor points which need to be addressed however.

Line 30 “amoebae” plural not “amoeba” singular.

Line 33. Perhaps “uncharacterized” would be better than “marginal” here as it is our current knowledge of the Rab GTPases that is marginal not them themselves?

Line 35. Yes, the current classification of genus *Acanthamoeba* is not complete or satisfactory, and it is almost certain that in due course a better nomenclature will be proposed after consultation across the interested parties. Whereas it seems unlikely that the species name *castellanii* would be retained, it is unhelpful at this stage to rename the Neff strain. Species name *terricola* has taxonomic problems also, so it's probably better to stick to the *Acanthamoeba castellanii* (Neff) description that the field understands until a complete and agreed system is established. Work on this problem is ongoing. Calling the Neff strain *terricola* and then having to change this again in a short time would cause unnecessary confusion.

Note that throughout the current manuscript the name *castellanii* is still being used in a few places (e.g. Lines 270, 722, 901, & 910).

More details about Scarlet are needed. It should be first of all stated that it is an RFP derivative. Other required details include is it pH sensitive?

Line 76. A small point but the references 7 and 8 are not a good match to support the argument that *Acanthamoeba* was adopted early as a model eukaryote. These references are dated 2013 and 2022? Instead I suggest the following early paper in which the now ubiquitous Myosin 1 motor protein was isolated from *Acanthamoeba* as an example of the utility of the amoeba as a eukaryotic model?

Pollard TD, Korn ED (1973) *Acanthamoeba* myosin I. isolation from *Acanthamoeba castellanii* of an enzyme similar to muscle myosin. J Biol Chem 248:4682–4690.

Line 116. “73% of the eleven Rab7 homologs”? Do the authors mean “8 of the 11 Rab7 homologs”?

Line 414. The “non-characterized, non-lytic, exocytic pathway” show similarities to the “vomocytosis” phenomenon (Seoane, P. I., & May, R. C. (2020). Vomocytosis: what we know so far. *Cellular Microbiology*, 22(2), e13145.). Have the authors considered this possibility?

Figure 4, 5 and 6. the panels showing the amoebae are labelled “bright field” but the images are clearly phase contrast?

Figure 5. It seems odd that the vacuoles containing *Stenotrophomonas maltophilia* labelled with pHrodo are uniformly red with no indication of the expected rod shape that the bacterium has? Is it possible that the label has become detached from the bacteria? Have the bacteria been digested? Is there another explanation? This apparent problem is also seen with the RFP scarlet expressing bacteria in Figure 7.

The expected rod shapes can be seen in Figure 7B but not actually in the cells or vesicles. This Part B of Figure 7 is very small, it contains important information and should be shown full size? The

fluorescence is used as a proxy for bacterial numbers. These experiments show that the bacteria continue to produce the RFP but can the authors make a strong claim that this reflects the actual intracellular growth of the bacteria? Rod-shaped Bacteria are clearly visible outside the amoebae but the vacuoles appear to be empty by phase contrast? (E.M. would sort this out but this may not be feasible for the authors to perform). This is important as this is the main message from the paper.

Response to Reviewers

Reviewer #1 (Comments for the Author):

Figure 3. Not sure why they haven't pseudo-colored the images. Images seem out of focus. For amoeba, a z-projection may have been better. No comments about the increased in vesicles in Rab7 overexpression?

R: We refrained from employing pseudo-coloration on the image because only a single channel (green) was utilized for all visual representations.

It is possible that images D and E, especially the former, may be slightly out of focus. Nevertheless, a portion of the blurriness can be attributed to photons emitted from sample planes above and below the focal plane, which contribute to the image's out-of-focus appearance and consequently reduce its resolution.

Although a z-projection would have been ideal, our use of a manual microscope (AxioVertA1, from Carl Zeiss) prevented the generation of a z-stack composed of multiple optical sections.

Despite these technical limitations, the images unequivocally demonstrate that the non-fused mEGFP signal is distributed throughout the cytoplasm and nucleus, whereas the mEGFP-Rab7a fusion protein localizes to vacuoles.

Fig 4 Missing arrow heads?

R: Yes, the reviewer is right. We added the missing arrow heads to the corrected figure.

Why not merge brightfield of EFH?

R: Due to the unsatisfactory outcome, we refrained from displaying a combined image of EFH. The merging of the three LCI channels proved to be challenging, as our method involved the use of a manual microscope, which resulted in the slight movement of vacuoles within the trophozoites during the process of manually changing filters for each individual image.

Are these scales right? Is scale different for H? It doesn't look like it matches?

R: The scales presented are accurate, as they were generated automatically by the ImageJ Quick-Figures plugin, utilizing the metadata contained within the native czi files produced by the microscope's zen-blue light software.

Figure 5. I get how annoying burning a scale in but I think you need to pick one and stick to it.

R: As previously mentioned, the scales presented are accurate, as they have been automatically generated by the ImageJ Quick-Figures plugin utilized for the creation of the multi-panel figures. Considering the diversity in trophozoite sizes and shapes, varying cropping regions have been employed, resulting in corresponding scale adjustments.

I'm struggling with it being a Toolbox when the transfection rate is 5-10%.

You are going to struggle to study anything related to the pathway or dynamics beyond a couple of images with that transfection rate. It is not a major improvement over what is currently in the literature.

R: Certainly, transfection efficiency is a critical factor for the success of downstream microscopic analyses. As indicated in the relevant sections of the manuscript, the low transfection efficiency presented a significant challenge. To address this issue, we employed FACS to enrich the population of fluorescent cells. It is worth noting that low transfection rates have also been reported in a recent publication (e.g., "Genetic manipulation of giant viruses and their host, *Acanthamoeba castellanii*," by Philippe N, Shukla A, Abergel C, Bisio H, published in Nature Protocols, 2023).

In the revised version, we have replaced the term "plasmid toolkit" with "plasmid set." While we believe that the former term was appropriate, given our development of four plasmids to generate both N- and C-terminal fusions of *Acanthamoeba* proteins to mEGFP or mCherry2, plus two mTn7 delivery plasmids to stably tag bacterial chromosomes, we acknowledge that some readers may expect a more comprehensive set of plasmids under the term "plasmid toolkit." Nonetheless, we stand by our contribution, as our plasmid set offers a significant advantage over existing methods, such as Bateman's plasmid pGAPDH-EGFP, which can only be used to generate fusions to the C-terminus of EGFP.

You've done a lot of plasmid work for both organisms here. Why that *S. maltophilia* specifically? would your plasmid work with others?

R: As previously discussed, we chose strain Sm18 for our research due to its status as one of the few known environmental isolates of *Stenotrophomonas maltophilia* sensu stricto (Vinuesa et al. 2018). We selected this particular strain from our extensive collection of environmental *Stenotrophomonas* samples as our model organism, as it displays greater ease in genetic manipulation, in part due to its slightly lower minimum inhibitory concentrations (MICs) for chloramphenicol and tetracycline, the only two commonly used antibiotics that work as useful selectable markers in our *S. maltophilia* strains.

In contrast to the recent plasmids developed by Mamat et al. (2023), our mTn7 plasmids have not been optimized for specific use with Sm18 or *Stenotrophomonas*. Consequently, these plasmids are expected to be functional in all Proteobacteria species susceptible to Tn7 transposition.

Also, using Neff (beyond the genomes that are available) is not super representative of the genus as a whole. Did you do anything to the Neff culture-wise to make it more "normal" versus an organism that has been axenically cultured for 60 years?

R: The culture was obtained from ATCC and cultured axenically as described in the Methods section using the standard PYG medium. Strain Neff is a legitimate *Acanthamoeba* strain that has been utilized in numerous recent relevant studies cited in the manuscript (e.g. Van der Henst et al. 2016 & 2018; Matthey-Doret et al. 2022; Bernard et al. 2022). At the inception of the project, the only annotated genome available in public databases was that of strain Neff. Considering that a phylogenomic analysis of *Acanthamoeba* Rab GTPases was a primary objective of this study, acquiring strain Neff was the most logical choice. Prior to initiating the genetic analysis, we confirmed that the cultures grew axenically in PYG, as well as grazing on live and dead *Escherichia coli* K12. The cultures were also

successfully induced to encyst and excyst. Based on these observations, we concluded that the strain was suitable for our study.

I don't know how much you extended the understanding of the Rab/Ras pathway in this paper beyond suggesting it co-localizes with intracellular bacteria. Has it really been established that *S. maltophilia* is only in acidified phagosomes?

R: Our study made a significant contribution to the field by demonstrating that a majority of the Rab7 homologues annotated in the *A. castellanii* Neff RefSeq genome are non-canonical, lacking key functional and structural motifs. Through phylogenomic analysis, we precisely identified the *Acanthamoeba* Rab7A ortholog and discovered that the RefSeq structural annotation for the gene was incorrect. Further analysis revealed that most of the other “non-canonical” Rabs predicted from the RefSeq genome sequence are likely to be wrong as well. This finding has important implications, including the possibility that a significant proportion of genes in the RefSeq genome contain misplaced introns and that the conclusions from previous phylogenomic and functional studies that included Neff proteins from RefSeq need to be interpreted with caution.

In our co-culture experiments, we cannot completely rule out the possibility that some viable *S. maltophilia* cells could remain extracellular despite using low MOIs. However, as outlined in the relevant Results, Discussion, and Methods sections, we did not observe Sm18 growth in the MMSalts-MOPS-Glc medium utilized, nor in 24h-old MMSalts-MOPS-Glc medium where *Acanthamoeba* had been grown (“conditioned medium”). Consequently, we are confident that replication occurs within phagosomes. However, as noted in the first paragraph of the Discussion, we observed that phagosomes in the Nef:mEGFP-Rab7A line were compromised, as long-term co-culture experiments (lasting more than 12 hours at MOIs greater than 25) resulted in the lysis of some vacuoles and the release of bacteria into the cytoplasm, leading to the lysis of trophozoites after 24 hours. Therefore, we conducted replication assays using the wild-type Neff strain, where we did not observe vacuole disruption. Additionally, unpublished results indicate that various Sm18 mutants replicate poorly or not at all in Neff phagosomes, further supporting our assertion that Sm18 replicates within phagosomes.

So it is pretty clear visually that *Acanthamoeba* is not super thrilled with mEGFP-Rab7A given the overwhelming amount of vesicles in the amoeba compared to mEGFP alone. I didn't see any discussion of that mentioned or the downstream effects of your experiment where the amoeba is clearly over producing vesicles and then you are making conclusions about the type of vesicles you are finding the bacteria in.

R: We would like to express our gratitude to the reviewer for bringing this issue to our attention, as it is a critical concern. In the revised version of the manuscript, the first paragraph of the discussion has been thoroughly revised to address this point, emphasizing the significant variation in the number and size of Rab7A-positive vacuoles present in different trophozoites by comparing, for instance, Figures 3, 4, and 5. This revised section highlights the heterogeneity in the data and provides additional insights into its potential source.

There's a bit of a cohesiveness issue with the paper. You made plasmids that labeled bacteria (again, not sure why a commercial plasmid wouldn't have worked; either way, most of that activity could be moved to the methods). *Acanthamoeba* you made plasmids, the transfection rate is very low (as is found with the other plasmids for amoeba so I'd be careful about suggesting some improvement). Your

plasmid clearly had a phenotypic impact that isn't addressed and needs to be. You found a bunch of Rab genes in Neff, picked one, and made a plasmid against it. You don't actually prove its an overexpression or that the protein is functional within amoeba, just that it is tagged and seems localized to the vesicle. This seems like a few papers together as one. I think you could focus on what you learned about bacteria infection in amoeba and maybe move some things into the methods rather than making it a bulk of the paper.

R: Upon commencing our project, we encountered the challenge of not finding mTn7 plasmids expressing mScarlet-I constitutively with CmR, the selective marker of choice for our MDR strain. Consequently, we were compelled to create a new plasmid that would serve as a valuable tool for both our current and future research. Mindful of its significance, we have made this plasmid available to the scientific community through AddGene.org. In response to the reviewer's suggestions, we have reorganized the manuscript by moving certain sections related to the Tn7 plasmids to the Methods section and excluding or condensing numerous sentences and paragraphs throughout the text. For instance, we have completely eliminated the paragraph describing *Stenotrophomonas* secretion systems and effectors at the end of the discussion.

This manuscript is quite expansive, as it encompasses extensive bioinformatics and thorough experimental approaches. We deem both methodologies indispensable for fully supporting our central finding: that *S. maltophilia* Sm18 replicates within acidified Rab7A-positive vacuoles of *A. castellanii* strain Neff. The revised version is more concise and better integrated.

We did not "arbitrarily analyze a plethora of Rab genes and select one to create plasmids against it," but rather focused our analysis specifically on Rab7 paralogs. Our functional analysis was concentrated on the phylogenetically and structurally validated *Acanthamoeba* Rab7A ortholog, which has been demonstrated to be a key marker of mature phagosomes in numerous organisms. Our study represents the first to examine this protein in *Acanthamoeba*-bacteria interactions. More precisely, we establish that the *S. maltophilia*-containing vacuole (SmCV) is an acidified, Rab7A-positive phagosome, indicating that this bacterium does not interfere with its initial maturation process and interacts extensively with the phagocytic pathway. These findings represent new insights into the *S. maltophilia/A. castellanii* interaction.

==

Reviewer #2

A. castellanii, reclassified as *A. terricola*, is a free-living amoeba with clinical and environmental significance. Despite being a valuable model organism, it is not as well-studied or utilised as other amoebae, such as *Dictyostelium discoideum* due to the limited availability of molecular biology tools. In this manuscript, Rivera et al. introduce a new set of plasmids that can be used to transfect *A. castellanii*, enabling the creation of fusions both upstream and downstream of fluorescent proteins. Using these plasmids, they were able to study the localisation of the GTPase ortholog Rab7A in *A. castellanii* and determine that it localises to acidified vacuoles. The authors also conducted infection assays with *S. maltophilia* and found that it is resistant to *A. castellanii* predation by forming a SmCV inside the amoeba, allowing it to replicate in this niche.

General comments:

I thoroughly enjoyed reading this manuscript and think it is very well-written. The Abstract and Importance sections were clear and concise, providing valuable information without being repetitive. The Introduction was succinct yet informative, including necessary background information and a thorough literature review.

While I appreciate the paragraph discussing the reclassification of *A. castellanii* to *A. terricola*, I do not believe it is relevant to this particular work. I am concerned that the new name in the title may cause confusion for some readers and potentially cause the article to miss its intended audience. I suggest acknowledging the reclassification and clarifying that *A. castellanii* will be used throughout the article for simplicity.

R: We are grateful to the reviewer for their suggestion to maintain the original classification of *A. castellanii*, which we have adhered to. Consequently, we have eliminated the paragraph discussing reclassification and replaced all instances of *A. terricola* with *A. castellanii* throughout the text.

It is evident that a great deal of work went into this manuscript and while I know that projects do not always/almost never proceed linearly, it appears to me that two separate stories are being merged together. Nonetheless, I still believe the manuscript is well-written, and I am unsure what changes could be made to improve it.

R: The manuscript is undoubtedly extensive and complex, yet we are resolute in our conviction that a comprehensive bioinformatics analysis and extensive experimental work are indispensable to unequivocally establish the primary conclusion of our research: that *Stenotrophomonas maltophilia* replicates within Rab7A-positive acidified vacuoles, thereby suggesting that the SmCV interacts substantially with the phagocytic pathway.

1. This is probably the obvious question, but why didn't you use the pGAPDH-EGFP as the backbone? As stated in the manuscript, it is the most common plasmid used to transfect *A. castellanii*, it's simpler to change one fluorescence gene than two promoters, and pGAPDH-EGFP also has an AmpR resistance gene for selection in *E. coli* when cloning. Regarding the cloning limitations of pGAPDH-EGFP, although the sequence is not available, it can be sequenced, and there are other tools to cut or insert fragments that don't require restriction sites/enzymes. Regarding promoter choice, Bisio et al. 2023 (<https://doi.org/10.1038/s41467-023-36145-4>) is an appropriate reference.

R: The primary reasons why we did not utilize pGAPDH-EGFP in our study are as follows:

1. Our request for the plasmid in 2021 was unsuccessful.
2. As indicated in the Results and Discussion sections of our project, a key objective was to establish a collection of plasmids to further research on the cellular microbiology of *Acanthamoeba*-bacteria interactions. Given that only the C-termini of *Acanthamoeba* proteins can be fused to EGFP in pGAPDH-EGFP and EGFP tends to form dimers, we believed that creating a set of plasmids with known sequences to fuse either the C- or N-termini of *Acanthamoeba* proteins to truly monomeric mEGFP and mCherry2 FPs would be a valuable contribution to the scientific community.

Thank you for making the plasmids available on Addgene, it is an important practice.

R: Certainly, we believe that this should be a standard practice for all published plasmids. We have added the AddGene accessions for the plasmids to Table 1 in the revised version.

2. Regarding the transfection of *A. castellanii*, did you use G418 to select for positive transformants and keep the line, or only FACS after transfection? How long did you keep the line for after transfection? If you passaged the culture after FACS, could you comment on plasmid/transfection stability without using selection?

R: We are grateful to the reviewer for raising this crucial aspect that was inadequately addressed in the previous version. In the "Generation and FACS enrichment of an *A. castellanii* line stably overexpressing an mEGFP-Rab7A fusion protein" section of the Methods, we have provided the necessary details. Furthermore, we briefly discuss stability concerns in the opening paragraph of the Discussion.

3. The phylogenetic analysis was thorough and well-explained, and the results were exciting. Kudos!

4. I found LCI and the two assays to characterise the localisation of Rab7A to be very fitting and creative. The micrographs were illustrative.

5. Transforming environmental strains is not a trivial task, so this is notable work, and I appreciate the creative use and adaptation of established protocols.

6. I'm curious, why did you perform the infections at 25°C instead of 30°C as in previous assays?

R: As stated in the Results section of the revised version on "***S. maltophilia* Sm18 establishes an intracellular replication niche in *A. castellanii* Neff**", long-term (> 24 h) co-culture experiments were performed at 25°C to avoid biofilm formation by Sm18 when incubated at this temperature.

7. The estimation of intracellular replication of *S. maltophilia* was an excellent addition.

Notes:

- Write the long form of "ppm" at least once.

R: thanks, this was done as suggested (see results section on "***Acanthamoeba castellanii* Neff trophozoites host *Stenotrophomonas maltophilia* Sm18 cells within acidified Rab7A-positive phagosomes.**") where we added post-primary contact (ppc).

- Figure 3 (C) legend: "firebrick" might not be understood by everyone as being a colour.

R: As per the recommendation, the term "firebrick" was replaced with "dark red" in legend for Figure 3 (C).

- Typos: Line 190 (Ran GTPase),

R: Ran GTPase is not a typo, but a distinct subfamily of small GTPases involved in nucleocytoplasmic transport, participating both in the import and export of proteins and RNAs from the nucleus. The two Ran proteins were utilized to root the phylogeny of Rab GTPases.

Line 309 (*A. terricola*/*S. maltophilia*) vs Line 495 (*Acanthamoeba-Stenotrophomonas*),

R: In the section titled "***Acanthamoeba castellanii* Neff trophozoites host *Stenotrophomonas maltophilia* Sm18 cells within acidified Rab7A-positive phagosomes**", we changed *A. castellanii*/*S. maltophilia* to *A. castellanii-S. maltophilia* for consistency.

Line 540 (GmR) changed to Gm^R

Line 853 (wright) corrected in the legend for Figure 1.

Line 925 (AlaphaFold2) corrected in the legend for Figure 3.

==

Reviewer #3

This study investigates the intracellular location of replication of an intracellular pathogen *Stenotrophomonas maltophilia* in *Acanthamoeba*, a facultative human pathogen. GFP- Rab7 GTPase fusion proteins are used to delineate organelle distribution. The work is of high quality and represents a significant step forward in the field and offers new tools to unlock the cell biology of *Acanthamoeba*. The paper is very well written and very clear. The main importance of this paper is the tools that it offers to other researchers, but the contents are extremely interesting also! There are a few relatively minor points which need to be addressed however.

Line 30 "amoebae" plural not "amoeba" singular.

R: Thank you for pointing out the error in the previous version. We appreciate your attention to detail.

Line 33. Perhaps "uncharacterized" would be better than "marginal" here as it is our current knowledge of the Rab GTPases that is marginal not them themselves?

R: We have revised the sentence to better reflect our current understanding of Rab GTPases. Instead of using "marginal," we have replaced it with "uncharacterized."

Line 35. Yes, the current classification of genus *Acanthamoeba* is not complete or satisfactory, and it is almost certain that in due course a better nomenclature will be proposed after consultation across the interested parties. Whereas it seems unlikely that the species name *castellanii* would be retained, it is unhelpful at this stage to rename the Neff strain. Species name *terricola* has taxonomic problems also, so it's probably better to stick to the *Acanthamoeba castellanii* (Neff) description that the field

understands until a complete and agreed system is established. Work on this problem is ongoing. Calling the Neff strain *terricola* and then having to change this again in a short time would cause unnecessary confusion.

R: We appreciate the reviewer's insightful recommendation to maintain the original classification of *Acanthamoeba castellanii*, which we have duly adopted. Consequently, we have removed the section discussing reclassification and replaced all instances of *A. terricola* with *A. castellanii* in the manuscript.

Note that throughout the current manuscript the name *castellanii* is still being used in a few places (e.g. Lines 270, 722, 901, & 910).

More details about Scarlet are needed. It should be first of all stated that it is an RFP derivative. Other required details include is it pH sensitive?

R: In the revised version of the Discussion, we present the following information to provide a general explanation for our choice of mScarlet-I: "We chose mScarlet-I (67) because it is reported to be a rapidly maturing, bright, acid-tolerant, and genuine monomeric red fluorescent protein derived from mRed7, a synthetic construct (101)."

Line 76. A small point but the references 7 and 8 are not a good match to support the argument that *Acanthamoeba* was adopted early as a model eukaryote. These references are dated 2013 and 2022? Instead I suggest the following early paper in which the now ubiquitous Myosin 1 motor protein was isolated from *Acanthamoeba* as an example of the utility of the amoeba as a eukaryotic model? Pollard TD, Korn ED (1973) *Acanthamoeba* myosin I. isolation from *Acanthamoeba castellanii* of an enzyme similar to muscle myosin. *J Biol Chem* 248:4682-4690.

R: As per the Reviewer's suggestion, we have revised the references and incorporated the seminal work by Pollard & Korn (1973) into our study.

8. Wetzel MG, Korn ED. 1969. Phagocytosis of latex beads by *Acanthamoeba castellanii* (Neff). 3. Isolation of the phagocytic vesicles and their membranes. *The Journal of cell biology* 43:90–104.

9. Byers TJ. 1986. Molecular biology of DNA in *Acanthamoeba*, *Amoeba*, *Entamoeba*, and *Naegleria*. *Int Rev Cytol* 99:311–341.

10. Pollard TD, Korn ED. 1973. *Acanthamoeba* myosin. I. Isolation from *Acanthamoeba castellanii* of an enzyme similar to muscle myosin. *J Biol Chem* 248:4682–4690.

Line 116. "73% of the eleven Rab7 homologs"? Do the authors mean "8 of the 11 Rab7 homologs"?

R: This was corrected.

Line 414. The "non-characterized, non-lytic, exocytic pathway" show similarities to the "vomocytosis"

phenomenon (Seoane, P. I., & May, R. C. (2020). Vomocytosis: what we know so far. Cellular Microbiology, 22(2), e13145.). Have the authors considered this possibility?

R: We appreciate the reviewer's input. In the Revised Discussion, we have written: "Nevertheless, the trophozoites remained intact and active, indicating that the bacteria were released through a non-lytic exocytic pathway, similar to the process observed in *Vibrio cholerae* (23) or the vomocytosis pathway described for *Cryptococcus neoformans* (105, 106)."

Figure 4, 5 and 6. the panels showing the amoebae are labelled "bright field" but the images are clearly phase contrast?

R: Although it is possible to discern a slight halo in the images, it should be noted that they were captured using an EC Plan-Neofluar 63x/1.25 Oil objective, which does not possess the capability for phase-contrast.

Figure 5. It seems odd that the vacuoles containing *Stenotrophomonas maltophilia* labelled with pHrodo are uniformly red with no indication of the expected rod shape that the bacterium has? Is it possible that the label has become detached from the bacteria? Have the bacteria been digested? Is there another explanation? This apparent problem is also seen with the RFP scarlet expressing bacteria in Figure 7.

R: It cannot be ruled out that a portion of the pHrodo label has become detached, preventing the observation of rod-shaped bacteria within the phagosomes. Additionally, the imaging was conducted using standard epifluorescence on a manual microscope, capturing only a single optical section. Therefore, it is likely that some of the blurriness is due to photons emitted from sample planes above and below the focal plane, which reduces the resolution. Moreover, it is possible that a portion of the cells within the phagosomes have suffered membrane damage and cell lysis, releasing some of the pHrodo label or RFP into the phagosome lumen. Future work should employ live cell imaging using a confocal microscope combined with transmission electron microscopy to determine the contributions of these factors and clarify the concerns raised by the reviewer.

The expected rod shapes can be seen in Figure 7B but not actually in the cells or vesicles. This Part B of Figure 7 is very small, it contains important information and should be shown full size? The fluorescence is used as a proxy for bacterial numbers. These experiments show that the bacteria continue to produce the RFP but can the authors make a strong claim that this reflects the actual intracellular growth of the bacteria? Rod-shaped Bacteria are clearly visible outside the amoebae but the vacuoles appear to be empty by phase contrast? (E.M. would sort this out but this may not be feasible for the authors to perform). This is important as this is the main message from the paper.

R: We have now expanded Figure 7B.

Notwithstanding the utilization of low MOIs, it is not feasible to entirely rule out the possibility that extracellular viable *S. maltophilia* cells may remain in our co-culture experiments. However, as stated in the pertinent Results, Discussion, and Methods sections, no growth of Sm18 was observed in the MMSalts-MOPS-Glc medium employed (which lacks L-methionine, a prerequisite for Sm18's growth, as it is an L-Met auxotroph), nor in 24h-old MMSalts-MOPS-Glc medium where *Acanthamoeba* had

been cultivated (“conditioned medium”). Consequently, we are rather confident that replication takes place within phagosomes. Nevertheless, as alluded to in the initial paragraph of the Discussion, it was observed that phagosomes in the Nef:mEGFP-Rab7A line were compromised, as prolonged co-culture experiments (lasting more than 12 hours at MOIs exceeding 25) resulted in the rupture of some vacuoles and the release of bacteria into the cytoplasm, which ultimately led to the lysis of trophozoites after 24 hours. Consequently, the replication assays were undertaken using the wild-type Nef strain, where no evidence of vacuole disruption was observed. Additionally, unpublished findings revealed that a variety of Sm18 mutants exhibit poor replication or no replication at all in Nef phagosomes, thereby further corroborating our assertion that Sm18 replicates within phagosomes.

Re: Spectrum02988-23R1 (**Phylogenomic, structural, and cell-biological analyses reveal that *Stenotrophomonas maltophilia* replicates in acidified Rab7A-positive vacuoles of *Acanthamoeba castellanii***)

Dear Prof. Pablo Vinuesa:

Your manuscript has been accepted, and I am forwarding it to the ASM production staff for publication. Your paper will first be checked to make sure all elements meet the technical requirements. ASM staff will contact you if anything needs to be revised before copyediting and production can begin. Otherwise, you will be notified when your proofs are ready to be viewed.

Sincerely,
Michael Ginger
Editor
Microbiology Spectrum